# Competition between transmission lineages mediated by human mobility shapes seasonal influenza epidemics in the US

Simon P. J. de Jong [1], Andrew J. K. Conlan [2], Alvin X. Han[1,3] & Colin A. Russell [1,3] ✉

Due to its climatic variability, complex mobility networks and geographic expanse, the United States represents a compelling setting to explore the transmission processes that lead to heterogeneous yearly seasonal influenza epidemics. By analyzing genomic and epidemiological data collected in the US from 2014 to 2023, we show that epidemics consisted of multiple co-circulating transmission lineages that could emerge from all regions and often rapidly expanded. Lineage spread was characterized by strong spatiotemporal hierarchies and lineage size correlated with timing of establishment in the US. Mechanistic epidemic simulations, supported by phylogeographic analyses, suggest that competition between lineages on a network of human mobility consistent with commuting flows drove lineage dynamics. Our results suggest that the processes that disseminate viruses nationwide are highly structured, but variability in the short-term processes that determine the locations, timing, and explosiveness of initial epidemic sparks limits predictability of regional and national epidemics.

In most countries, seasonal influenza viruses cause periodic epidemics, leading to substantial morbidity and mortality. However, key epidemic characteristics such as size, timing, and viral composition vary substantially from year to year[1–3]. The recurrent nature of influenza virus epidemics arises from a complex co-evolution between virus and host immunity[4–6], which acts in concert with factors such as climate[1,7–9], multi-scale human mobility[10–16], and human contact patterns[17] to shape disease transmission potential. Knowing how these factors jointly shape the transmission processes leading to yearly epidemics is essential for a predictive understanding of epidemic dynamics[18,19]. This predictive understanding is a key public health target[20], and substantial efforts are dedicated to forecasting the timing of epidemic onset and epidemic peaks, as well as the viral composition of epidemics, to aid public health planning[3,21–26].

The US forms a compelling setting to explore fundamental questions about the drivers of seasonal influenza virus transmission due to its geographic expanse, climatic variability and complex mobility networks. Analyses of sequence data have shown that a highly dynamic global viral metapopulation, driven by complex patterns of inter-regional viral migration, re-seeds US epidemics each year[27]. In turn, these epidemics can go on to seed epidemics elsewhere[13,27]. However, the processes that turn initial epidemic seeds in the US, introduced from abroad, into full-fledged epidemics remain poorly understood. For example, do epidemics comprise a single epidemic wave that sweeps across country, or rather consist of many distinct co-circulating transmission lineages[28,29]? Similarly, are epidemic waves the result of gradual expansion of early-season transmission chains, or the result of explosive spread when conditions become favorable for rapid transmission? Further questions remain regarding the mobility drivers of viral spread, such as the roles of air travel and commuting in disseminating viruses nationwide[12,16,30–32].

[1]Department of Medical Microbiology & Infection Prevention, Amsterdam University Medical Centers, University of Amsterdam, Amsterdam, The Netherlands. [2]Department of Veterinary Medicine, University of Cambridge, Cambridge, UK. [3]These authors contributed equally: Alvin X. Han, Colin A. Russell. ✉e-mail: c.a.russell@amsterdamumc.nl

Most prior studies into seasonal influenza epidemic dynamics in the US have relied primarily on virological and syndromic surveillance data, such as pneumonia and influenza (P&I) mortality data or influenza-like illness (ILI) data[12,16,30]. However, such data cannot effectively distinguish between distinct chains of transmission, limiting the precision with which the underlying dynamics of epidemic emergence, establishment and viral dissemination can be reconstructed[19,30]. Hence, we turned to genomic data, collected during routine surveillance in the United States. By decomposing epidemics into contributions of distinct transmission lineages, and reconstructing their individual spread, we aimed to gain more fine-grained insight into the processes that turn initial epidemic sparks into widespread national epidemics.

## Results

### Influenza epidemics consist of distinct co-circulating transmission lineages

First, we characterized the transmission lineage structure of US seasonal influenza epidemics, to investigate whether they tend to comprise many distinct co-circulating transmission lineages that independently emerged in different states, or rather consist of a single dominant transmission lineage that propagates across the country. We analyzed 30,508 whole-genome seasonal influenza virus sequences from the 48 contiguous states and the District of Columbia, collected during routine surveillance from 2014 to 2023. In this period, all four influenza A subtypes and influenza B lineages (henceforth, subtypes) caused epidemic activity, but patterns of subtype dominance differed from season to season (Supplementary Fig. 1). We phylogenetically grouped these viruses into clusters that exhibit a comb-like branching structure, suggestive of exponential spread[33]. Given the exponential nature of influenza epidemics, we posit that groups of viruses with such a rapidly expanding branching structure plausibly descended from a single ancestral virus in the United States that was introduced from abroad (Fig. 1a, Supplementary Fig. 2–5).

Using this procedure, we clustered 81.2% of sequences into 3842 lineages of at least two viruses. In most seasons, a relatively small number of transmission lineages accounted for the bulk of sequenced viruses (Fig. 1b), with median 5 lineages (range 1–13) cumulatively accounting for ≥50% of sequences across subtypes and seasons. Transmission lineage diversity differed substantially across seasons (Fig. 1b): in the 2015/2016 A/H1N1pdm09 epidemic, a single transmission lineage accounted for >50% of sequences, normalized across states, whereas in the 2016/2017 A/H3N2 season the largest lineage accounted for only 6.4%. Lineage structure was present for both circulating influenza A virus subtypes and both influenza B virus lineages, though transmission lineage clustering results are likely more error-prone for influenza B viruses given their lower evolutionary rate[34], particularly in seasons that saw relatively little circulation and were less densely sampled. We found no relationship between transmission lineage diversity and metrics of epidemic timing (Pearson $r = -0.16$, $P = 0.63$) or intensity (Pearson $r = -0.12$, $P = 0.73$). Only a small proportion of lineages spread widely across the country (Fig. 1c): among the 1,104 identified transmission lineages that accounted for at least 5% of sequences in a season in at least one state, 144 (13.0%) lineages did so in at least 10 states, and 27 (2.4%) did so in at least 25. These results indicate that in most seasons, seasonal influenza epidemics are the result of the co-circulation of multiple independent chains of transmission, consistent with previous studies into individual seasons at the national and state level[28,29]. These conclusions are robust to the choice of parameters for the procedure used to cluster viruses into transmission lineages (Supplementary Fig. 6).

### Lineage size correlates with timing of establishment of epidemic activity

We hypothesized that the first lineages to emerge in any season, for a given subtype, would be larger. Here, we defined lineage size as the proportion of sequences of the corresponding subtype that a lineage

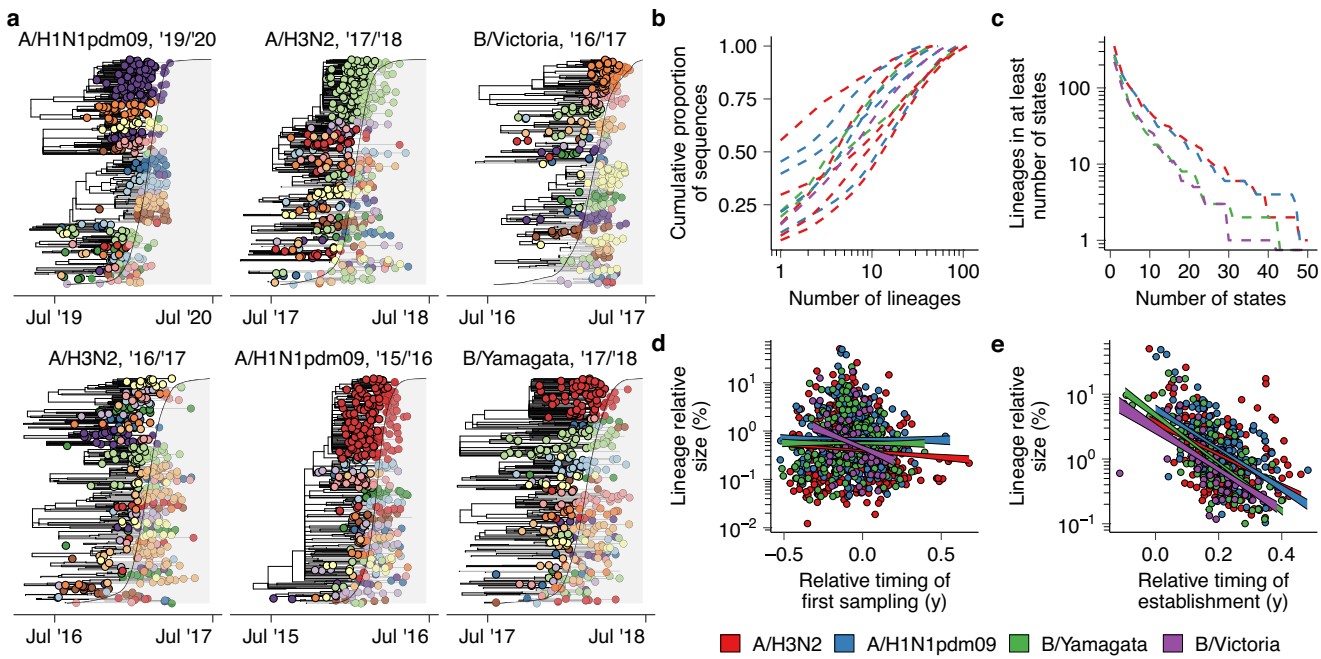

**Fig. 1 | Lineage structure of US seasonal influenza epidemics. a** Example phylogenetic trees with tips colored by transmission lineage. The shaded gray area corresponds to the cumulative proportion of the season's nationwide positive tests in public health laboratories of the corresponding subtype that had been collected at each point in time. **b** Lineage size distribution by season and subtype. Each line represents the cumulative proportion of sequences that is accounted for by a number of lineages on the x-axis. **c** The number of subtype-specific lineages that accounted for >5% of sequences in a season-subtype in at least the number of states

on the x-axis. **d** Relationship between the first collection date of virus in a lineage and the lineage's nationwide size normalized by state ($n = 1030$). Lineage sampling dates were computed relative to the timing of nationwide epidemic onset, which was defined as the first week in which >5% of the season's cumulative positive tests had been collected. Lines correspond to 50% CI given linear fit. **e** Relationship between the timing of establishment of substantial circulation of a lineage and its nationwide size ($n = 569$). Lineage establishment timing was computed relative to nationwide epidemic onset analogous to (**d**).

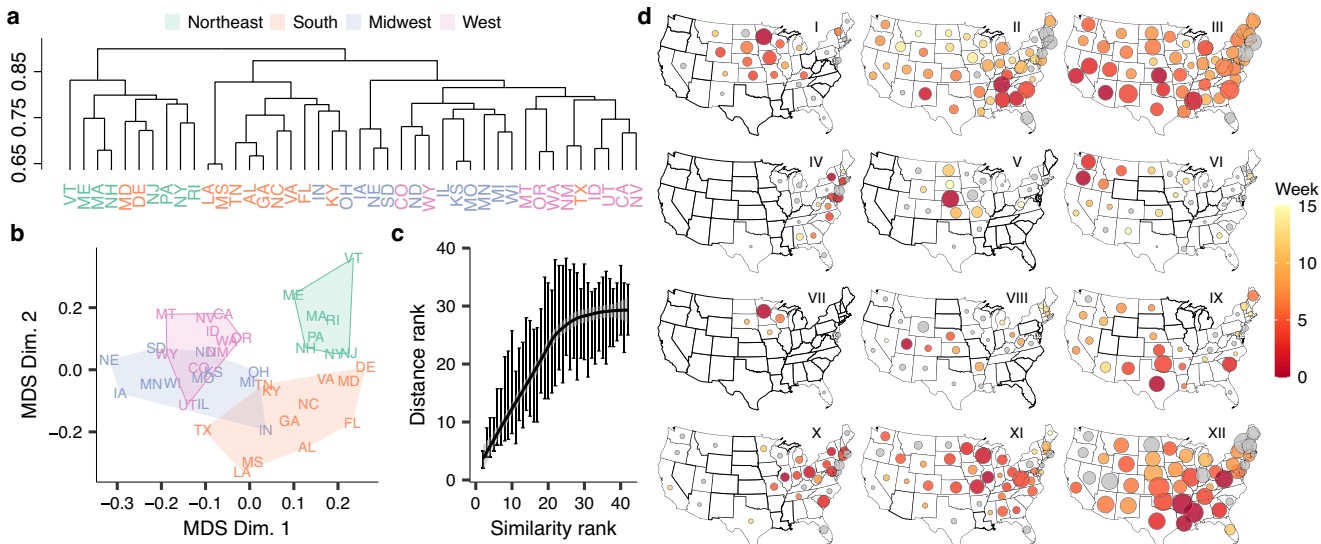

**Fig. 2 | Spatial structure of US seasonal influenza epidemics. a** Complete-linkage hierarchical clustering of pairwise state-to-state transmission lineage composition Bray-Curtis similarities across all subtypes, colored by census region. **b** Multi-dimensional scaling plot of the pairwise lineage composition Bray-Curtis similarity among states, colored by census region. **c** Relationship between pairwise transmission lineage compositional similarity rank and pairwise centroid distance rank ($n = 42$). Vertical lines show 50% CI of distance rank for each value of rank similarity; line corresponds to LOESS fit to medians. **d** Examples of lineage spatiotemporal spread for a geographically and temporally representative set of lineages. In each map, circle size and color correspond to the relative size and establishment timing of the corresponding lineage in each state, respectively. Gray fill corresponds to unknown lineage establishment timing.

accounts for in a season across all states, where each state has an equal weight. However, across all subtypes and seasons, the relationship between time of first sampling of a lineage and (log) lineage size was weak (Spearman $\rho = -0.08$, $P = 0.012$) (Fig. 1d). Some transmission lineages that were first sampled a prolonged interval prior to onset of the national epidemic proliferated into peak periods (Fig. 1d). However, many of the most successful lineages were first sampled close in time to national epidemic onset (Fig. 1d). For example, by the time of first sampling of the largest lineage in the highly severe[35] 2017/2018 A/H3N2 season, >10% of all the season's sequences had already been collected, but the lineage accounted for >40% of sequences around the epidemic peak following rapid expansion (Fig. 1a). The fact that a transmission lineage could rapidly sweep to national dominance despite emerging at a time when many other transmission lineages were already circulating suggests that influenza epidemic dynamics are strongly influenced by heterogeneous short-term epidemiological processes.

These results suggest that early-season transmission chains frequently go extinct before the onset of substantial epidemic activity. However, we hypothesized that if a lineage did cause substantial epidemic activity early on, it would likely be successful nationwide. We defined the timing of lineage establishment as the first week in which the lineage accounted for substantial epidemic activity in at least one state. When defining lineage establishment week as the first week a lineage had cumulatively accounted for >5% of the state's total epidemic activity of that subtype, establishment week correlated strongly with nationwide lineage size for all subtypes (Spearman $\rho = -0.46$, −0.50, −0.70, −0.71 for A/H3N2, A/H1N1pdm09, B/Yam, B/Vic, respectively, $P < 0.001$ for all) (Fig. 1e). Statistical models indicated each week's delay in establishment resulted in a decrease in lineage size of 16.9%, 15.1%, 14.6%, and 13.0% for A/H3N2, A/H1N1pdm09, B/Yam, and B/Vic, respectively ($P < 0.001$ for all). The fact that the lineages that first established substantial epidemic activity somewhere in the US were more likely to be successful suggests that the states with the earliest epidemic onset have outsized contributions to which viruses will circulate nationwide. The proportion of the earliest-establishing transmission chains that resulted in a successful transmission lineage

decreased as the cumulative incidence threshold used to determine establishment week decreased (Supplementary Fig. 7). This suggests that while a first-mover advantage shapes lineage success, substantial levels of epidemic activity are necessary for a lineage's early circulation to be predictive of substantial nationwide spread.

## Transmission lineages are highly spatially structured

To quantify spatial structure in the circulation of transmission lineages, we computed the Bray-Curtis dissimilarity of transmission lineage compositions across states. Here, lower dissimilarity is indicative of more frequent circulation of the same lineages. Qualitatively, hierarchical clustering of the resulting similarity matrix recapitulated the geography of the United States, with relatively higher similarity for states within the same census region (Fig. 2a, Supplementary Fig. 8), suggestive of substantial spatial structure. Projecting the similarities among states onto a two-dimensional surface further recapitulated this spatial structure; for example, states belonging to the Northeast and Southeast appeared to form distinct clusters (Fig. 2b). However, the continuous distribution of states on the plane suggests states cannot consistently be classified into distinct communities, suggestive of substantial inter-regional mixing.

Across all seasons and subtypes, epidemics in states in closer geographic proximity more frequently comprised the same transmission lineages (Mantel test, $P < 0.001$) (Fig. 2c), with the highest similarities found for adjoining states (highest: MS-LA, MO-KS, GA-AL, NH-MA, UT-ID). Stratifying by season and subtype, this correlation between distance and similarity was present in most, but not all, subtypes that accounted for >20% of nationwide positive tests in their respective season (Supplementary Fig. 9). Visualizing lineages' spatial distribution and proliferation across states revealed a striking landscape of spatiotemporal spread. Individual lineages emerged from all regions of the US, each with distinct spatial signatures (Fig. 2d). Across seasons and subtypes, many lineages exhibited a radial spatiotemporal progression and were highly regional (e.g. I, V, VI, VII, Fig. 2d), whereas other lineages saw rapid cross-country spread (e.g. II, III, XII, Fig. 2d). Together, these results show that at the transmission lineage level, US seasonal influenza epidemics are highly spatially structured.

## Consistent source-sink dynamics are absent across seasons and subtypes

Next, we reconstructed where in the United States successful transmission lineages initially established. US source-sink dynamics have been the subject of debate, with studies suggesting that the South represents a dominant source region[16,30]. Using BEAST[36], we identified the Health & Human Services (HHS) region that represented the most likely region of initial expansion for each of the 262 transmission lineages that accounted for >0.5% of sampled viruses, normalized by state, in their respective season. To ensure our results were robust to sequence sampling effects, we performed these analyses with a population-weighted subsampling strategy and a subsampling strategy with a uniform number of sequences across regions. We found that the origin region of successful lineages differed substantially from season to season. Of the seven transmission lineages that accounted for >10% of sequences in a single season, normalized across states, three likely first established in the southern US (HHS regions 4 and 6; e.g., lineages II and XII, Fig. 2d), one likely emerged in the western US (HHS region 9; lineage III, Fig. 2d), one in the midwestern US (HHS regions 7 and 5; lineage XI, Fig. 2d), one in the northeastern US (HHS region 1), and one could not consistently be attributed to a single region across both sampling strategies.

To investigate if the dominant origin region of sampled viruses differed among states, we computed origin profiles that quantify, for each state, what proportion of viruses belonged to transmission lineages that initially expanded in each of the HHS regions. These profiles differed substantially across states (Supplementary Figs. 10, 11). For example, averaged across both subsampling strategies, the proportion of sequences that corresponded to transmission lineages initially expanding in HHS region 4, encompassing most of the southeastern US, ranged from 39.3% in South Carolina (HHS region 4) and 27.4% in Arkansas (HHS region 6) to 13.0% in Arizona (HHS region 8). Similarly, lineages initially expanding in HHS region 10, encompassing the Pacific Northwest, accounted for 15.3% of sampled viruses in Idaho (HHS region 10), 10.7% in North Dakota (HHS region 8), and only 2.7% in Arkansas (HHS region 6). States in closer geographic proximity saw more similar origin profiles, even if they corresponded to different HHS regions (Mantel test, $P < 0.001$). Across all states, a relatively limited proportion of viruses corresponded to lineages that originated from the state's own HHS region (median 17.3%, range 11.4–29.8% for uniform subsampling strategy), suggesting a high degree of viral mixing at the national level. Importantly, origin profiles were strongly correlated across the two subsampling strategies (Spearman $\rho = 0.81$, $P < 0.001$). Together, these results suggest that influenza virus source-sink dynamics are highly heterogeneous, without consistent source regions of successful lineages, but are spatially structured.

## Mechanistic simulations suggest commuting flows correlate with viral spread

Our analyses established a strong correlation between timing of lineage establishment and lineage size (Fig. 1e). However, this correlation does not account for a substantial portion of the observed variation in transmission lineage size. We hypothesized that differences among states in their connectedness through human mobility, coupled to competition between lineages for susceptible hosts, could account for further variation. In this hypothesis, lineages that established in more connected states have a greater potential to rapidly spread to other states, thereby outcompeting lineages that emerged around the same time in less connected states. This would explain why some lineages spread widely following local establishment, whereas other lineages remained highly spatially constrained (Fig. 2d).

This hypothesis appears to explain lineage dynamics in the 2018/2019 A/H3N2 season. The beginning of this season was dominated by A/H1N1pdm09 viruses, but it also saw the rapid expansion of A/H3N2 viruses that were associated with reduced vaccine effectiveness[37].

Phylogenetic analyses, integrated with epidemiological data, indicate that A/H3N2 circulation in this season was dominated by two swiftly establishing lineages, appearing to emerge from Georgia (GA, lineage 1) and Nebraska (NE, lineage 2), respectively (Fig. 3, left). Spread of the Nebraskan lineage was regional and radial, causing substantial epidemic activity mainly in the immediately surrounding states (Fig. 3, left). Conversely, the lineage from Georgia quickly spread to almost all states with a less prominent temporal hierarchy, although it appeared to arrive in neighboring states first. We hypothesized that competition from the Georgian lineage explained why spread of the Nebraskan lineage remained so regional. In turn, this would explain why the Georgian lineage failed to spread substantially in Nebraska and immediately surrounding states. We used the reconstructed spread of these lineages as a ground truth to address long-standing questions regarding the underlying mobility determinants of influenza virus spread.

To test the hypothesis that competitive interactions between lineages, coupled to mobility, drive lineage spread, we explored if we could reproduce the spread of these two lineages with mechanistic epidemic simulations. We used a metapopulation model to simulate the inter-state spread of multiple co-circulating lineages that compete for disease-susceptible hosts with perfect cross-immunity in a susceptible-infected-recovered (SIR) epidemic framework. We initialized the simulations in the lineages' respective establishment weeks, in their respective onset states (Georgia and Nebraska) and simulated forward in time to model the spread of the two lineages. To ascertain the predominant mobility drivers of viral spread, we parameterized rates of state-to-state mobility using either commuting flows, extracted from the US Census Bureau, or air travel data, extracted from the US Department of Transportation.

When using commuting flows to parameterize rates of inter-state travel, we could reproduce the observed spread of the two lineages with striking accuracy: the simulations recapitulated the radial spread from Nebraska, and the relative success of the lineage emerging from Georgia (Fig. 3, left). The model of inter-lineage competition also explains why the lineage from Georgia failed to cause epidemic activity in Nebraska and the immediately surrounding states. The correspondence to the observed distribution of lineages across states was much poorer ($\Delta\ell = -81.93$) when using air travel flows, with an absence of substantial spread from Nebraska. Furthermore, simulated lineage establishment timings better matched observed establishment timings when using commuting data (MSE = 6.0 wk) than when using air travel data (MSE = 11.7 wk) Together, these mechanistic simulations suggest that commuting flows are a strong correlate of viral spread and suggest that competitive interactions between lineages, mediated by mobility flows, shape the distribution of lineages across states.

## Spatial segregation and limited competition allow lineages from small states to spread widely

Lineage dynamics in the 2018/2019 A/H3N2 season are consistent with the conjectured gravity-like spread of seasonal influenza viruses[12], with a lineage originating from a populous, highly connected state (in this case, Georgia, population ~11 million) spreading quickly through strong long-range connections, while spread from a smaller state (Nebraska, population ~2 million) was slower and more local. In this model, Georgia's high degree of connectivity and earlier onset allowed lineage 1 to spread to other states more rapidly than lineage 2. Nevertheless, the substantial spread of the Nebraskan lineage shows that a lineage emerging from a small state can proliferate, even if it co-circulates with a lineage emerging from a more populous state, if it sees sufficiently early establishment and is spatially segregated from the lineage emerging from the larger state.

Our results suggest that by facilitating spread from less populous states, the short-range spatial coupling reflected in commuting flows is

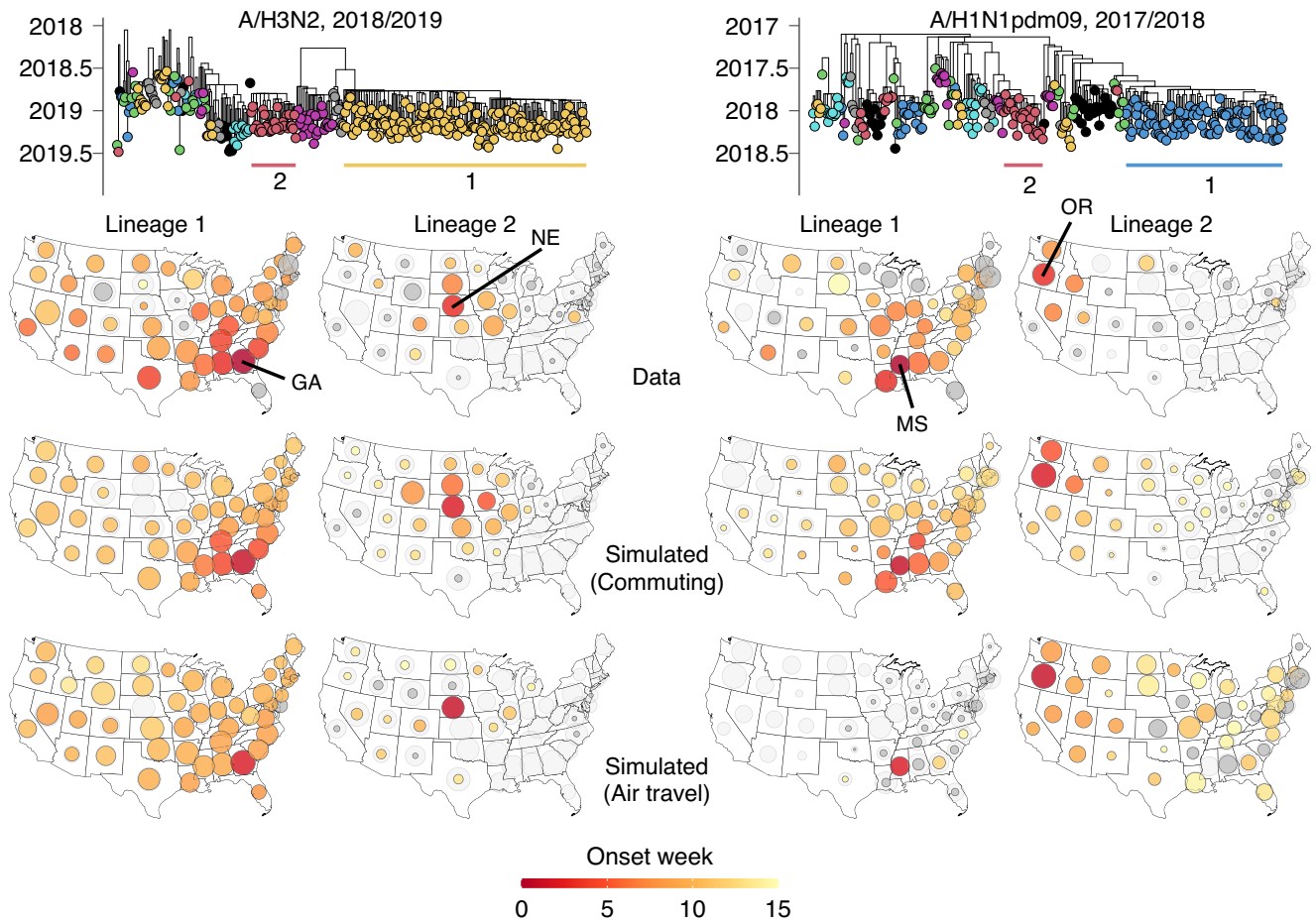

**Fig. 3 | Mobility drivers of influenza virus spread.** Phylogenies represent the 2018/2019 A/H3N2 season and 2017/2018 A/H1N1pdm09 season, with the two largest lineages labeled in each. For both seasons, the maps in the top row visualize the reconstructed spread of the labeled lineages, with size and color corresponding to lineage size and establishment timing, respectively. Middle and bottom maps show simulated spread of the two lineages for each of the two seasons, using commuting and air travel data, respectively. Light gray circles represent the total proportion of sequences in a state that was accounted for by simulated lineages. In each state, circle sizes for simulated lineages are scaled such that the sum of simulated lineages' sizes is proportional to the proportion of sequences accounted for by the simulated lineages (i.e., the light gray area). Dark gray fill corresponds to absence of an establishment week (for top row, potentially due to missing data), or establishment after the 15th week.

a key determinant of seasonal influenza virus spread. This notion is further supported in the 2017/2018 A/H1N1pdm09 season, in which the two largest lineages appeared to emerge in Mississippi (MS) and Oregon (OR), respectively (Fig. 3, right). When using commuting flows, the relative degree of spread of the two lineages could be reproduced. Despite its relatively small population, Mississippi's high connectivity through commuting flows allowed lineage 1 to rapidly spread beyond local constraints. In contrast, due to Oregon's relatively limited connectivity and the later establishment of lineage 2, competition from lineage 1 likely constrained the spread of those viruses to the Western United States. When using only air travel to parameterize inter-state mobility, the simulations overestimated the degree of spread from Oregon relative to Mississippi, with too slow spread from Mississippi, compared to the ground truth, resulting in poorer fit ($\Delta\ell = -1.14$ for observed sample counts, $\Delta MSE = 5.5$ for observed establishment timings) (Fig. 3, right).

Using counterfactual simulations, we explored how mobility interacts with establishment timing to shape the spread of individual lineages. Simulations indicate that had lineage 2 established in Nebraska four weeks later (with lineage 1's establishment timing unchanged), stronger competition would have constrained lineage 2 to Nebraska and immediately adjacent states. Conversely, if it had established four weeks sooner, lineage 2 would have been approximately equal in size to lineage 1, with earlier onset compensating for lower connectivity (Supplementary Fig. 12a). Similarly, later onset for lineage 2 in the 2017/2018 A/H1N1pdm09 season would have constrained it to the Pacific Northwest, whereas earlier onset would have facilitated substantially more expansive spread (Supplementary Fig. 12b).

## Inter-lineage competition explains differences in lineages' spread

To further test the hypothesis that competition between lineages on the human mobility network explains individual lineages' spread, we performed in-depth investigations into the 2019/2020 B/Victoria season, which was characterized by anomalously high levels of epidemic activity[38]. The lineage composition of this season was highly spatially diverse, with the largest lineages appearing to originate in or in the vicinity of California (lineage 1), Florida (lineage 2), Texas (lineage 3), Louisiana (lineage 4), Nevada (lineage 5), and Washington (lineage 6), respectively (Fig. 4). Lineage dynamics in this season highlight the heterogeneity of processes of lineage establishment. For example, the lineage emerging from Florida likely emerged in the spring of 2019 (posterior mean TMRCA May 5, 95% CrI March 14–June 12), seemingly persisting throughout the 2019 summer in Florida, potentially providing a counterexample to the general trend that viruses do not

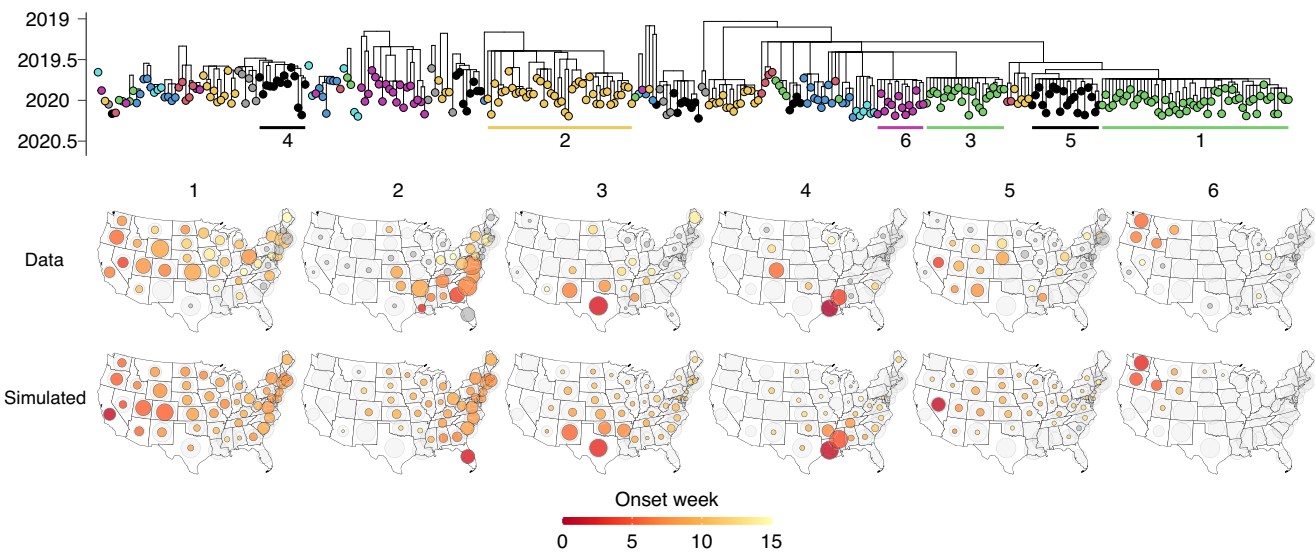

**Fig. 4 | Mobility-induced competition drives lineage spread.** Phylogeny represents the 2019/2020 B/Victoria season, with the six largest lineages labeled in order of size. Top row of maps represents the reconstructed spread and distribution of each of the six largest lineages. Bottom row of maps represents the simulated spread and distribution of the six lineages, initialized in the lineages' likely onset state and onset week, using a combination of air travel data and commuting data. Circle sizes are scaled as in Fig. 3.

persist between seasons[28]. Conversely, the lineages from California, Nevada, and Texas spread widely following rapid establishment, despite much later emergence (e.g., lineage 1: posterior mean TMRCA August 29, 95% CrI July 3–September 25, Fig. 4). Some lineages spread to over half of all states (e.g., lineages 1 and 2 from Florida and California, respectively), whereas other lineages were more spatially constrained (Fig. 4).

Using a combination of commuting flows and air travel flows, the simulations reproduced the spread of individual lineages and their distribution across states (Fig. 4). Differences in mobility flows, in combination with competition for susceptible hosts, parsimoniously explain why the lineages emerging from California and Florida spread widely, whereas the lineages from Louisiana and Washington were more spatially constrained. Commuting flows in isolation also yielded a strong fit, but underestimated spread from Nevada, suggesting that residual air travel flows not captured by commuting could play a role in viral dissemination (Supplementary Fig. 13). Simulations using only air travel deviated from the ground truth particularly by underestimating short-range viral migration from Louisiana and Washington, and provided a worse fit to the observed distribution of lineages across states ($\ell = -347.7, -393.2, -400.0$ for fit to lineage sample counts for combined, commuting, and air travel models, respectively) and the relative timing of lineage establishment in different states (MSE = 7.6, 7.9, 16.6, respectively, for the three models) (Supplementary Fig. 13).

Our metapopulation simulations demonstrate a strong model fit while directly leveraging data on mobility flows without fitting any mobility-related parameters, indicating these mobility metrics parsimoniously explain observed dynamics. To derive a more general quantification of the gravity-like nature of viral spread, we fit a model where the propensity $p_{ij}$ for an individual from state $j$ to travel to state $i$ (conditional on traveling) depends on state $i$'s population size $N_i$ and the distance between states $d_{ij}$ as $p_{ij} \propto N_i^\tau / d_{ij}^\rho$. In this model, the daily probability for an individual from a given state to travel was informed by the commuting and air travel data. This general model accurately recapitulated observed lineage dynamics, with a strong fit to sample counts ($\ell = -329.7$), but with moderately reduced fit to establishment timing (MSE = 9.5) (Supplementary Fig. 13). We estimated a rapid decay with distance ($\rho = 1.4$, 95% CI 1.2–1.5) with a statistically significant

dependence on destination population size ($\tau = 0.5$, 95% CI 0.3 − 0.8) (Supplementary Fig. 14). However, if we assumed the daily probability of traveling was constant across states, the model yielded a substantially worse fit than the models directly informed by commuting and air travel data ($\ell = -429.6$, MSE = 11.2). This indicates that accounting for differences among states in outward travel rates is necessary to capture salient features of influenza virus spread.

## Viral genomes suggest viral migration correlates with commuting

The mechanistic simulations suggest that commuting flows are the primary mobility drivers of influenza virus spread. However, these analyses could only be performed for the seasons with relatively low lineage diversity, as the large number of co-circulating lineages in some seasons rendered sample counts too low for individual lineages to yield a reliable ground truth for reconstructions of spread. We sought to confirm that commuting is the predominant correlate of viral spread across the full set transmission lineages in the study period. Hence, we probed whether the same mobility processes were reflected in the genetic relationships among viruses belonging to the same transmission lineage.

Using phylogeographic methods, we reconstructed the migration history of the sampled viruses in each transmission lineage, tracing the state-to-state jumps between the lineage root and each virus. Using these reconstructions, we quantified the relative viral jump contribution $x \to y$, representing the proportion of reconstructed migration events to and from state $y$ that was accounted for by state $x$. We found that the states with the greatest jump contribution to another state tended to be the states that were most strongly connected through commuting flows to that state (Spearman $\rho = 0.63$, $P < 0.001$) (Fig. 5a). The correlation with air travel was weaker (Spearman $\rho = 0.32$, $P < 0.001$) (Fig. 5b), as was the correlation with state centroid distance (Spearman $\rho = -0.39$, $P < 0.001$). This provides further support for the notion that commuting is a strong correlate of the mobility processes that disseminate seasonal influenza viruses across the United States.

The highest values of the relative jump contribution $x \to y$ were found when state $x$ was highly populous and state $y$ was in close geographic proximity to state $x$. For example, the highest values across all

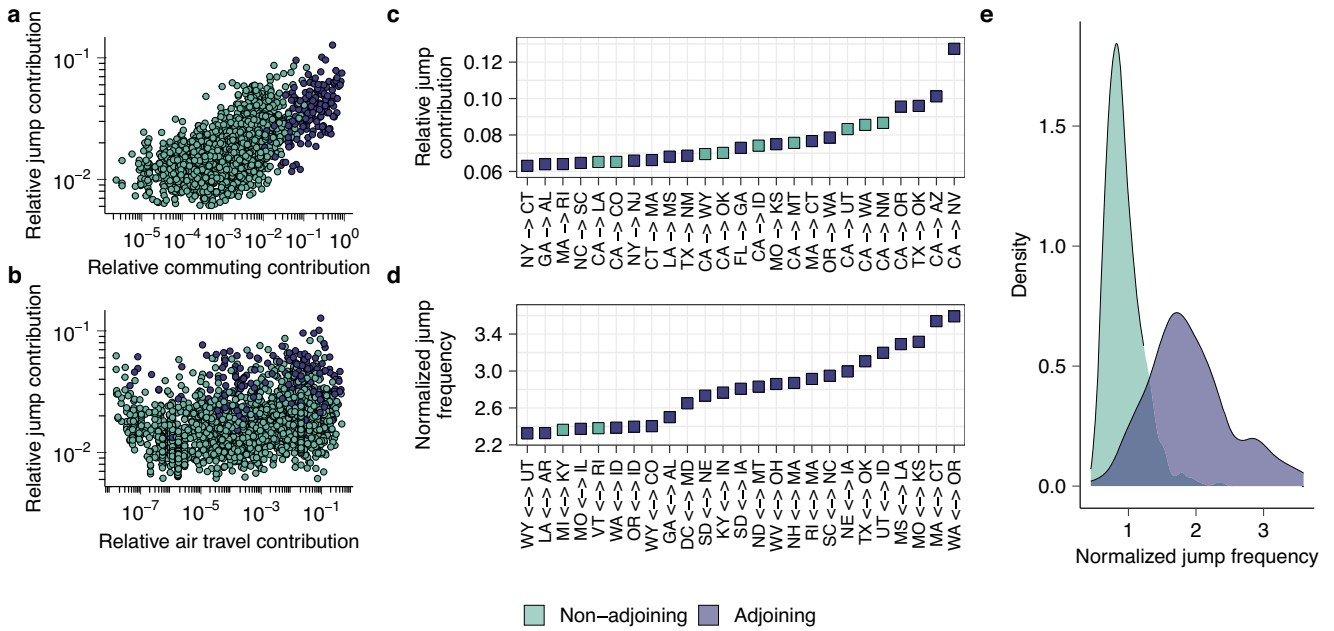

**Fig. 5 | Phylogeographic analyses of mobility drivers. a** Relationship between the relative contribution of each other state to a state's inbound and outbound reconstructed viral migration events, and the other state's relative role as a state's commuting destination (*n* = 2352). **b** Analogous to (**a**), for air travel data (*n* = 2352).

**c** Visualization of the 25 highest values of the relative jump contribution.
**d** Visualization of the 25 highest values for the normalized pairwise jump frequency.
**e** The distribution of normalized pairwise migration frequencies for pairs of adjoining and non-adjoining states.

pairs were found for CA → NV, CA → AZ, TX → OK, CA → OR, and CA → NM (Fig. 5c). This is expected under classical gravity-like spread where, for any given state, the highest connectivity is expected to be to states that are in close geographic proximity and highly populous. However, this pattern could be confounded by higher sample counts for the most populous states, as higher sample counts in a spatial grouping in phylogeographic analyses will a priori be expected to lead to more reconstructed migration events even in the absence of any spatial signal in the data. Hence, we computed an alternative metric, the normalized pairwise jump frequency $x \leftrightarrow y$, which represents the proportion of migration events to/from state $y$ that is accounted for by state $x$, normalized relative to the mean proportion of migration events that state $x$ accounts for across all states. The highest values were for adjacent states that are strongly connected through commuting flows (highest: WA ↔ OR, MA ↔ CT, MO ↔ KS, MS ↔ LA, UT ↔ ID) (Fig. 5d), and this value was substantially greater for adjacent than non-adjacent states (Fig. 5e). This result suggests that when accounting for effects of population size and/or sampling, viral migration is strongly skewed toward short distances, further supporting the important role of short-range spatial coupling. To identify differences among subtypes in the degree of distance-dependence of viral spread, we computed the correlation between centroid distance and pairwise jump contribution for each season and subtype individually. While we established differences among subtypes and seasons, these differences were not consistently associated with subtype (Supplementary Fig. 15).

## Discussion

By integrating epidemiological and genomic data, our analyses reveal the structure of seasonal influenza epidemics at the resolution of individual transmission lineages, which co-circulate and together constitute national epidemics. The lineage structure of epidemics cannot reliably be identified from epidemiological data alone, but is key to understanding the spatial dynamics of influenza; for example, patterns of inter-state spatial coupling identified in this study differ substantially from patterns extracted solely from influenza-like illness

data[24]. Furthermore, the strong spatial coupling between more populous states previously identified from epidemic synchrony could be the result of concurrent but independent processes of epidemic establishment, rather than the result of hierarchical spread between states[12,31]. Our results indicate that competition for susceptible hosts between lineages establishing in different states induces a strong spatial structure in seasonal influenza epidemics and is an important determinant of which viruses will circulate where.

Lineage structure is also key to understanding seasonal influenza virus source-sink dynamics. For example, our inferences of source-sink dynamics differ from those of studies that implicitly assumed a single lineage comprised all epidemic activity in a season[39]. Previous studies based on ILI data have posited that the South represents the dominant source of influenza epidemic waves[16,30]. Our analyses reveal the frequent early establishment and national success of lineages emerging in the South. However, this pattern was not consistent across seasons, and our results demonstrate a striking diversity and season-to-season variability in the location of emergence of successful transmission lineages. A key question is what drives this variability. Potential explanations include year-to-year differences among regions in environmental factors that shape early-season transmission potential[40], or differences among regions in susceptibility to circulating viruses, possibly related to patterns of viral circulation in prior seasons[2,41]. Alternatively, differences among states in the supply of epidemic seeds from abroad could lead to differences in the locations of initial lineage establishment. Such effects could arise from among-state differences in the preferred destinations for travelers[12], in combination with year-to-year variability in where substantial circulation of viruses with high fitness occurs globally[13,14,27,42] in the period when epidemic seeds would find fertile ground for transmission in the US. Testing this hypothesis and leveraging the predictive value that may be gained for prediction of epidemic composition underscores the need for strong global genomic and epidemiological surveillance networks.

Our results suggest that viral migration is well-described by commuting flows, which generate the network on which co-circulating lineages compete for disease-susceptible individuals. Commuting data

has previously been suggested to drive influenza virus spread based on analyses of ILI data[12,30], but this has not been shown mechanistically or validated against phylogenetically supported instances of viral spread across (sub)types[11,19,30]. While we found a clear dominance of commuting over air travel when considering these metrics in isolation, our results also suggest that air travel flows not captured in commuter surveys could play a role in viral dissemination. Our metapopulation analyses provide a framework to fit models of human mobility to epidemic pathogens in the presence of lineage structure, which could be used to evaluate more refined spatial models and their parameterizations in future work. However, it is striking that we could reproduce lineage spread dynamics using mechanistic simulations when parameterizing mobility directly using commuter surveys, without fitting any mobility-related parameters. This indicates that commuting flows provide a parsimonious explanation for observed patterns of viral spread. We could only perform our analyses at the state level owing to that being the level of spatial resolution in most sequence metadata, and analyses at other spatial scales may yield different results regarding modes of virus spread[30,43,44]. Furthermore, our results do not rule out potential roles for alternative types of mobility, such as movement of children not well-captured by commuting flows[45].

Our results have implications for disease surveillance. For example, the prominent spatial hierarchies of lineage spread provide an explanation why including spatial coupling has been found to increase forecasting performance[23,24]. Early-season virological surveillance data have been shown to give clues as to epidemic subtype composition[25], but the fact that the earliest-sampled viruses will often not propagate into periods of peak epidemic activity, but rather lineages that rapidly expanded at later times often dominate, potentially limits the predictive utility and specificity of early-season genomic surveillance efforts. On the other hand, the strong correspondence between lineage establishment timing and lineage size suggests a potentially important role for nowcasting efforts to identify the locations of early epidemic establishment which, when combined with high-resolution genomic surveillance in those areas, could be leveraged to generate more robust predictions of lineage spread[24].

Our analyses have a number of limitations. The procedure used to classify viruses into transmission lineages could introduce errors, but sensitivity analyses suggest that the conclusions of our study are robust to such misspecification. Our analyses sought to identify differences among subtypes and seasons in their dynamics of viral spread. Mobility flows underlying the spread of influenza B viruses are potentially different from those for influenza A viruses as a result of differences in the age distribution of infection[27]. Furthermore, viral spread has been suggested to be particularly localized in seasons that saw the circulation of a novel antigenic variant[30]. While we identified differences among individual seasons and subtypes in their spatial structure and the distance-dependence of viral spread, we are cautious about direct comparison across seasons and subtypes. For example, the lower evolutionary rate of influenza B viruses could result in reduced detectable spatial signal, as the increased time between successive mutations would lead to an attenuated evolutionary imprint of the viral migration processes. If a lineage spreads particularly quickly, the spatial signal could be similarly obscured. While our analyses are unable to confidently ascertain differences among subtypes, our mechanistic simulations were able to recapitulate observed patterns of spread using commuting data for influenza A and B viruses, suggesting broadly similar mechanisms drive the spread of both.

Our results indicate that competition for susceptible individuals is a primary determinant of lineage success: lineages that establish earlier and in better-connected states are on average larger. Our simulations assume all individuals are equally susceptible to all lineages, but differences in antigenic properties among lineages could potentially explain further variation in lineage success. For

example, US influenza seasons where circulating viruses saw more mutations in antigenic sites of the hemagglutinin (HA) and neuraminidase (NA) surface proteins were associated with larger epidemic size[41], and the viral composition of the 2017/2018 US A/H3N2 season was consistent with dominance of the viruses to which population immunity as measured using standard serological assays was lowest[46]. This hypothesis is also consistent with, for example, the dominance of the antigenically drifted lineage in the 2018/2019 A/H3N2 season[37]. However, our study is underpowered to detect fitness differences between lineages that could be attributed to antigenic change, due to the study's relatively noisy data and limited time period, confounding due to orthogonal determinants of lineage fitness (such as environmental conditions in the state of emergence), combined with the complexity of influenza virus antigenic dynamics[4,6].

An essential question is what drives the heterogeneity in the speed of lineage establishment. We found that many of the most successful transmission lineages emerged very shortly before nationwide epidemic onset and established rapidly, sometimes sweeping to national dominance despite substantial competition from other contemporaneous transmission chains. In some seasons the outcompeted contemporaneous lineages were highly genetically related to the lineage that would come to dominate, with a common ancestor in or after the summer preceding the corresponding season. Importantly, this suggests that viral factors such as antigenic novelty are unable to fully explain the heterogeneity in the speed of lineage establishment. These differences could be related to potentially highly stochastic environmental factors or differences in human contact patterns. The heterogeneity of transmission processes raises important questions regarding the predictability of early-season seasonal influenza epidemiological dynamics at multi-week time horizons, even in the presence of perfect data. The fact that seasonal influenza forecasts rarely outperform models based on historical baseline activity at timescales greater than a few weeks[18,47] is likely tied to these heterogeneities. Even if our results shed light on the potentially predictable underlying drivers of viral migration, disentangling the roles of the various environmental, host, and viral factors is likely necessary to probe the limits of seasonal influenza epidemic predictability.

## Methods
### Data
We downloaded all genome sequences for A/H3N2, A/H1N1pdm09, B/Victoria, and B/Yamagata viruses (all four referred to as 'subtypes' throughout) collected from humans in the United States between July 1st 2014 and July 1st 2023 with all eight gene segments available from the GISAID[48] EpiFlu database. We retained only virus sequences with the US Centers for Disease Control as submitting laboratory to minimize the impact of targeted sequencing investigations that are potentially not representative and could bias the data, yielding a dataset consisting of 30,508 viruses (A/H3N2: 14,235, A/H1N1pdm09: 8,155, B/Yamagata: 3,543, B/Victoria: 4,584). We downloaded weekly proportions reporting for influenza-like illness, weekly counts of positive influenza A and B tests in clinical laboratories and weekly counts of positive tests by subtype in public health laboratories by state and season, and the number of positive tests in public health laboratories at the national level, from the CDC FluView website (https://gis.cdc.gov/grasp/fluview/fluportaldashboard.html). Data on commuting flows, stratified by origin and destination county, for 2016–2020 was extracted from the US Census Bureau (https://www.census.gov/data/tables/2020/demo/metro-micro/commuting-flows-2020.html). Data on air travel fluxes between states, stratified by origin and destination airport for the year 2017 was extracted from the US Bureau of Transportation Statistics (https://www.transtats.bts.gov/DL_SelectFields.aspx?gnoyr_VQ=GED&QO_fu146_anzr=).

## Phylogenetic analyses

We aligned the sequences for each gene segment and subtype using MAFFT[49]. To identify molecular clock outliers, we clustered the sequences for each segment and subtype individually using CD-HIT[50] with a clustering threshold of 99.5% nucleotide identity. Using a single representative virus for each cluster, we built a phylogenetic tree for each subtype and segment using FastTree[51], fit a molecular clock to each tree using TempEst[52], and removed 40 sequences belonging to clusters for which the representative virus was classified as a molecular clock outlier. We constructed whole-genome maximum-likelihood phylogenies using the cleaned dataset. To minimize risk of bias in phylogenetic inference resulting from tree incongruence due to gene reassortment, we inferred whole-genome phylogenies separately for groups of viruses that were highly genetically related across all segments. To identify these groups of viruses, we constructed phylogenies for each segment individually using all viruses for the entire period using IQTree[53] with a HKY[54] substitution model. We clustered these trees by identifying groups of viruses where, for each taxon within that group, there was at least one other taxon in the group that saw a patristic distance to the former taxon that was smaller than a given distance threshold. We defined this distance threshold as the expected number of mutations over a two-year period given the estimated molecular clock rate for that segment and subtype, using a more relaxed three-year period for the MP and NS segments for additional lenience given their lower evolutionary rate. Using these segment-specific clusters, we assigned each taxon a genome-wide cluster as the combination of the taxon's segment-specific clusters. These genome-wide clusters were identified separately for each season, spanning the period from January 1st of the preceding winter period up to July 1st of the following year. For each of the genome-wide clusters, we constructed whole-genome phylogenies in IQTree[53] using a HKY[54] substitution model with a segment-proportional model to account for varying evolutionary rates across segments[55]. Given these maximum-likelihood phylogenies, we constructed temporally resolved trees using TreeTime[56] using a fixed clock rate estimated using TempEst.

## Transmission lineage identification

We used a modified version of Phydelity[57] to delineate the whole-genome time trees for each of the genome-wide clusters into individual transmission lineages. We identified groups of highly related viruses for which the branching structure follows the comb-like shape expected under an exponential growth population dynamic process, where most coalescent events happen close in time to the common ancestor. Specifically, we required that for each transmission lineage, a certain proportion $p$ of all coalescent events must occur within a particular period $t$ of the putative lineage's root, with $p$ and $t$ specified. Given these constraints, Phydelity aims to cluster as many taxa as possible. If $t$ is very high and $p$ is very low the constraints imposed on the tree shape of a transmission lineage are relatively less stringent, potentially resulting in erroneous clustering if genetically similar viruses were independently introduced, but a very stringent threshold (i.e., low $t$ and high $p$) might erroneously discard true transmission lineages. To balance sensitivity and specificity we chose $p = 10\%$ and $t = 1$ month in the main text. We performed sensitivity analyses for $p = 5\%$, 15%, and 20%. We visualized the clustered trees using ggtree[58]. There was a strong log-linear relationship between a state's population size and its sequencing rate relative to its population size (Pearson $r = -0.73$, $P < 0.001$), but some states had substantially greater sampling rates than would be expected under the identified relationship given their population size. To ensure representative sequence counts for each state, we subsampled the taxa in each state, for each season-subtype pair, such that no state had a number of sequences more than 0.5 log units greater than the regression-predicted number given its population size.

## Lineage spread reconstruction

To reconstruct the spatiotemporal spread dynamics of individual lineages, we integrated the sequence sampling dates with influenza-like illness and virological surveillance data. For each season and state individually, we computed the influenza type-specific disease signal by multiplying the proportion reporting influenza-like illness in each week by the proportion of tests positive for influenza A and B separately, yielding a measure of type-specific incidence ('ILI + '), consistent with previous work[15,25,41]. Following previous work[59], we applied a 4253H, Twice smoother, implemented in the *sleekts* R package, to smooth the epidemic curves, in order to increase robustness of estimation of lineage establishment weeks. To extract transmission lineage-specific epidemic curves, we fitted the sampling dates of taxa belonging to individual transmission lineages to the reconstructed ILI+ curves. Specifically, for each state and type (i.e., influenza A or B) individually, we computed the contribution of each taxon in each week using a normal density with mean equal to the taxon's sampling date, with a two-week standard deviation. In each week, the contribution of a transmission lineage to epidemic activity in a state was given by the sum of taxon contributions in that week from viruses belonging to the transmission lineage in question, divided by the sum of taxon contributions in that week across all viruses. The weekly relative incidence of a transmission lineage was then computed by multiplying the transmission lineage's contribution in that week by that week's ILI+ signal for the corresponding type. We retained taxa that were not assigned to any transmission lineage as a separate group ('unclustered'). For each transmission lineage, we computed lineage size by dividing the number of taxa sampled in each state belonging to a particular lineage by the total number of sequenced viruses in the state for that subtype in the corresponding season; we computed lineage establishment timing as the first week the lineage had accounted for at least 5% of total incidence in that season (if at all), using the mapping of sequence sampling dates to incidence data described above. We computed nationwide lineage size as the sum of state-specific relative sizes for a given subtype divided by the number of states; hence, each state was equal-weighted, irrespective of the state's population size or sample count. We defined nationwide time of lineage establishment as the first week of lineage establishment in any state. We used a linear mixed-effect model with a random effect for season, fitted for each subtype individually, to estimate the effect of establishment week on lineage size. We visualized lineage spread using the *usmap* R package.

## Spatial structure analyses

To characterize the similarity of lineage compositions across all pairs of states, we sampled 20 clustered viruses from each state for each season across all subtypes (or retained all if fewer than 20 sequences were available) and computed the Bray-Curtis similarity of the transmission lineages corresponding to the sampled viruses using the *vegdist* command in the *vegan*[60] R package. We performed this procedure 50 times, retaining the mean value of each pair of states' similarity across all replicates. All analyses were performed for the seasons from 2014/2015 to 2019/2020 and 2022/2023, omitting the 2021/2022 season due to its aberrant epidemic dynamics following the COVID-19 pandemic; this season saw substantial levels of circulation during the summer period, complicating the delineation of lineages into individual seasons. We retained only states with at least 10 sequences in all seasons, leaving a set of 42 states. We performed hierarchical clustering on the similarity matrix across all seasons and subtypes using the *hclust* R function, using complete linkage clustering. We performed isometric multi-dimensional scaling using the *isoMDS* function in the *MASS* R package. To compute the correlation among states between compositional similarity and centroid distance, we correlated the similarity matrix with states' centroid distances using the *mantel* command in the *vegan* package. We

performed these analyses at the individual season-subtype pair level in the same fashion, sampling 10 viruses from the set of clustered taxa sampled in that season for that subtype and computing the Bray-Curtis similarity as described above.

## Source-sink phylogeographic inference

For the analyses of source-sink dynamics, we performed phylogeographic analyses in BEAST[36] for all transmission lineages that accounted for at least 0.5% of all sequenced viruses in a given season across all subtypes. We performed these analyses at the level of Health and Human Services (HHS) region, to allow for substantial spatial granularity while also having sufficient sequence counts per spatial unit. We used Thorney BEAST, implemented in BEAST[36] v1.10.5, to estimate a distribution of time-resolved phylogenies for each individual lineage, marginalizing over bifurcating topologies consistent with the potentially multifurcating input tree. We used divergence trees estimated with IQTREE v2.0.3 as input trees, extracting the subtrees that corresponded to each transmission lineage. We furnished all transmission lineages with fewer than 50 taxa with an exponential growth coalescent prior, and all transmission lineages with at least 50 taxa with a Skygrid[61] prior. We estimated a single clock rate for each season–subtype pair. For each season–subtype pair, we ran a single MCMC chain for 500 million iterations, sampling lineage trees every 5 million states. We assessed convergence using Tracer[62], and generated a set of 90 posterior trees for each transmission lineage using TreeAnnotator (https://beast.community/treeannotator), removing the first 10% as burn-in.

We performed discrete trait phylogeographic inference[63] using the posterior lineage trees. We used a CTMC model for migration where we assumed equal rates of migration between all regions, as many lineages had relatively few sequences and tree topologies were relatively uninformative, prohibiting reliable estimation of pairwise migration rates between regions. We ran these analyses for 100 million iterations, sampling every million, and removed the first 10% as burn-in. We leveraged stochastic mapping[64] to identify migration events on the posterior phylogenies. Using these reconstructed migration events, we identified the likely origin of each lineage in each posterior sample as the HHS region that was the source for most of the first 10 migration events. We used this definition for the lineage origin rather than simply the reconstructed root region as we were primarily interested in the rapid expansion of lineages, and the root could be affected by the inclusion of unrelated singleton viruses in the analysis that did not contribute to lineage expansion.

Using the lineage origin posterior distributions, we then computed state-specific origin profiles which represent the posterior proportion of sampled viruses in a focal state that belonged to lineages that were reconstructed to have originally expanded in each HHS region, aggregating across all subtypes and seasons and weighting each lineage according to the total proportion of circulation it accounted for in the corresponding season across all subtypes in the focal state. To investigate spatial structure in these source profiles, we correlated the Euclidean distance of these profiles between states with the centroid distance between the states using a Mantel test. Because we expected higher similarity between states in the same HHS region because they represent a single group in the phylogeographic reconstructions, we only performed this analysis for states that were not in the same HHS region. For any given state, we only included those seasons where that state had at least 10 sampled viruses when computing the source profiles, to prevent stochastic sampling effects from biasing results when sequence counts were low.

To minimize bias resulting from differences in sampling rates among the geographical groupings, we used two subsampling strategies for these analyses. For the first subsampling strategy, sequences from states that had a sequence count that was greater than expected from the regression line relating sequencing rate to population size

were subsampled to the sequence count predicted from the regression line given the state's population size. Hence, the sample count for each HHS region was roughly proportional to the region's population size. For the second sampling strategy, we ensured that the number of taxa included for each HHS region was approximately uniform, irrespective of the HHS region's population size. For each season-subtype combination, we computed the sequence count for this strategy as the 25th quantile of the number of sequences in each HHS region in the population-proportional subsampling scheme used above. For regions with more sequences than this value, sequences were randomly subsampled.

## Metapopulation model

To reproduce transmission lineage spread we used a mechanistic epidemic model that simulates the inter-state spread of co-circulating pathogens that compete for disease-susceptible individuals with perfect cross-immunity, following SIR epidemic dynamics. To limit the computational burden, we used a deterministic model that stratifies each epidemiological state into further compartments for individuals remaining in their home state and for those visiting another state. The model dynamics are then as follows:

$$\frac{dS_{ij}}{dt} = -\beta \sum_v I_{vij} \frac{\sum_j S_{ij}}{\sum_j N_{ij}} + S_{jj} l_{ij} - S_{ij} r$$

$$\frac{dS_{ii}}{dt} = -\beta \sum_v I_{vii} \frac{\sum_j S_{ij}}{\sum_j N_{ij}} - \sum_j S_{ii} l_{ji} + \sum_j S_{ji} r$$

$$\frac{dI_{vij}}{dt} = \beta I_{vij} \frac{\sum_j S_{ij}}{\sum_j N_{ij}} - \gamma I_{vij} + I_{vjj} l_{ij} - I_{vij} r \tag{1}$$

$$\frac{dI_{vii}}{dt} = \beta I_{vii} \frac{\sum_j S_{ij}}{\sum_j N_{ij}} - \gamma I_{vii} - \sum_j I_{vii} l_{ji} + \sum_j I_{vji} r$$

$S_{ij}$ represents the number of susceptible individuals originating from state $j$ that are currently in state $i$ and $I_{vij}$ represents the number of individuals originating from state $j$ that are currently in state $i$ and are infected by a virus corresponding to transmission lineage $v$. $l_{ij}$ represents the outward travel rate from $j$ to $i$ and $r$ represents the return rate. We assumed $r = 1\,d^{-1}$, recovery rate $\gamma = 4\,d^{-1}$, and $R_O = 1.35$[12], with transmission rate $\beta = R_O \times \gamma$.

For the commuting analyses, we computed the number of commuters between each pair of states by aggregating across origin and destination counties in each state. We symmetrized these counts (even though the raw data was already highly symmetric, $r = 0.9998$) and computed the daily travel rate $l_{ij}$ as the number of symmetrized commuters between states $i$ and $j$ divided by the population size of state $j$. For air travel analyses, we computed the rate of air travel between states as the symmetrized total number of passengers in 2016 between airports located in the two states. We computed the daily travel rate $l_{ij}$ as the yearly trip count divided by 365 and the population size of state $j$. For the simulations that used a combination of air travel and commuting flows, $l_{ij}$ was defined the maximum of the pairwise air travel and commuting rates to account for the possibility that some of the commuting flows were accounted for by the air travel data. The model was implemented in C++, interfacing with R using Rcpp[65].

## Mechanistic simulations

To simulate competitive spread of lineages, we initialized each lineage in its first week of establishment in its likely onset state and simulated forward in time. We used the reconstructed first week of establishment in any state, rather than in the onset state, to account for situations where incidence data was absent for the likely onset state. In the simulations for each mobility modality, we allowed each lineage's onset week to vary to up to two weeks after or two weeks before its estimated onset week, to account for error in the estimation of establishment timing with noisy data and situations where no incidence data was available for the putative index state. For each

individual mobility modality, we retained the configuration with the highest likelihood. Lineages were initialized with an infected population of $1\times10^{-5}$ times the index state's population size. Analogous to the ground truth reconstructions, we computed the size of each lineage in the reconstructions as the proportion of infections across simulated lineages in a state that was attributable to a particular lineage and the week of establishment as the first week a lineage had caused >5% of total infections across simulated lineages. We computed the log-likelihood ($\ell$) for each simulation from the product across all states of the multinomial probability of the observed sample counts for the transmission lineages given the relative proportions of infections attributable to the respective lineages in the simulations. We further quantified model fit with the mean squared error (MSE) of simulated vs. estimated lineage establishment timings. To fit the gravity model, we assumed the total daily rate of movement from each state was either 1) given by the sum of the daily outflows computed from the commuting data and the air travel data for that state, or 2) constant across states, given by the median of the state-specific rates computed in (1). We assumed the best-fit lineage onset weeks as estimated for the simulations using a combination of commuting and air travel. Given the total rate of movement from state $j$, we computed the flux to state $i$ as $p_{ij} \propto N_i^\tau / d_{ij}^\rho$. We simulated for $\rho$ from −0.5 to 3 and $\tau$ from −0.5 to 2, each in increments of 0.1. We computed the 95% CI based on profile likelihoods given the simulations for each combination of $\rho$ and $\tau$.

### Phylogeographic correlates of mobility

To correlate mobility with rates of inter-state viral migration as reflected in the lineage phylogenies, we performed Bayesian phylogeographic analyses at the state level using the population-weighted subsampling strategy as described above. For each pair of states $x$ and $y$, we computed the relative jump contribution $x \to y$ as the proportion of reconstructed migration events (Markov jumps) to and from state $y$ that was accounted for by state $x$. We analogously computed the proportion of travelers from state $y$ that had state $x$ as destination for the air travel and commuting data and correlated these quantities with the relative jump contributions as estimated from the phylogenies. Here, we added a pseudocount for pairs of states with zero commuters or air travelers. We also computed the normalized relative jump frequency $x \leftrightarrow y$, which represents the proportion of migration events to/from state $y$ that is accounted for by state $x$, normalized relative to the mean proportion of migration events that state $x$ accounts for across all states. These values are highly symmetric (Pearson $r = 0.997$), and hence we symmetrized to subsume pairs of states. By comparing the jump frequency between states relative to the states' mean, this metric is not prone to potential biases resulting from differences in sampling across states. However, a limitation of this metric is that only allows for ascertainment of the effect of distance and not of characteristics that are intrinsic to a single location, such as population size.

### Reporting summary

Further information on research design is available in the Nature Portfolio Reporting Summary linked to this article.

## Data availability

Sequence data is available from GISAID. GISAID Accession Numbers are provided in Supplementary Data 1. Epidemiological data is available from CDC FluView (https://gis.cdc.gov/grasp/fluview/fluportaldashboard.html). Commuting data is available from the US Census Bureau (https://www.census.gov/data/tables/2020/demo/metro-micro/commuting-flows-2020.html). Air transportation data is available from the US Department of Transportation (https://www.transtats.bts.gov/DL_SelectFields.aspx?gnoyr_VQ=GED&QO_fu146_anzr=).

## Code availability

Computer code underlying the results of this paper is available at https://github.com/AMC-LAEB/usa_flu and https://doi.org/10.5281/zenodo.15211496.

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

## Acknowledgements

We gratefully acknowledge all data contributors, i.e., the Authors and their Originating laboratories responsible for obtaining the specimens, and their Submitting laboratories for generating the genetic sequence and metadata and sharing via the GISAID Initiative, on which this

research is based. GISAID Accession Numbers are provided in Supplementary Data 1. This work was supported by US National Institutes of Health grant R01AI132362 (S.P.J.d.J., A.J.K.C., C.A.R.) and the Netherlands Organization for Scientific Research (NWO) Veni grant 9150162210121 (A.X.H.).

## Author contributions

Conceptualization: S.P.J.d.J., and C.A.R. Methodology: S.P.J.d.J., A.J.K.C., and A.X.H. Investigation: S.P.J.d.J. Visualization: S.P.J.d.J. Supervision: A.X.H., C.A.R. Writing—original draft: S.P.J.d.J. Writing—review & editing: S.P.J.d.J., A.J.K.C., A.X.H., and C.A.R.

## Competing interests

The authors declare no competing interests.
