## [Peer Review file · Nature Communications]

Competition between transmission lineages mediated by human mobility shapes seasonal influenza epidemics in the US

Corresponding Author: Mr Simon de Jong

Version 0:

Reviewer comments:

Reviewer #1

(Remarks to the Author)

Review of Jong et al. Nature Communications

In this article, de Jong and colleagues use a genomic centric approach to characterize Influenza outbreaks in the US. By combing influenza genomic data with a variety of other available data sources, including ILL-clinical data reported by the CDC and census-level commuter data, the authors were able to determine that influenza outbreaks are variable in size and lineage composition by year and region in the US, and lineage spread and competition is influenced by commuting patterns. The authors present a robust and well-analyzed dataset that tackles a standing question in influenza virus epidemiology: are epidemics in the US defined by single lineages, multiple co-circulating lineages, and does this vary from year to year and from state to state? In my opinion, the real challenge of a study like this is to develop a reasonable way to define transmission lineages, and I think the authors present a logical approach to defining influenza transmission lineages that is repeatable appropriately caveated. Collectively, this is an interesting manuscript that answers important questions using a robustly sampled dataset. I only have a few questions and points of clarification, listed below.

Line 546: Have the authors determined how representative the sequencing dataset is in relation to the case burden caused by influenza sub-types? There are >30,000 genomes included in this study and I am curious if those are representative of the yearly incidence of influenza cases caused by each subtype, which I think is a straight forward question to answer. Given how much the authors determine transmission lineage and extrapolate lineage-specific incidence, it would be important to determine if some influenza infections are over represented in the sequencing data.

Line 580: The authors state each sequenced virus segment was assigned a segment-specific cluster identity. The authors then assigned a genome-wide cluster identity as the combination of individual segment identities. Did all the segment-specific identities need to be identical for viruses to be assigned the same genome-wide cluster identity? If not, how much variability was allowed? I am wondering if this approach would be able to account for potential recombination events, or if recombination would inherently result in a different genome-wide cluster identity.

Line 642: Did estimates of lineage-specific incidence match the clinical ILL type-specific incidence from year to year? I think this comparison would also be a way to determine if some influenza types are over- or under-represented in the genomics dataset compared to true incidence caused by each influenza subtype.

Line 656: I believe a word is missing in this sentence.

Using this dataset, would it be feasible to assess the year-to-year composition of transmission lineages to specifically ask the question if any lineages are maintained between seasons? While this seems unlikely due to influenza seasonality, but I wonder the same transmission lineages are ever identified in multiple seasons, presumably because it has maintained transmission globally.

(Remarks on code availability)

Reviewer #2

(Remarks to the Author)

The authors sought to characterize transmission in outbreaks of seasonal influenza within the United States. It is challenging to study seasonal influenza dynamics as outbreaks are often comprised of many concurrent chains of transmission, especially from the perspective of larger spatial scales. The authors utilize genomic epidemiology and mathematical modeling techniques to characterize “epidemic emergence, establishment and viral dissemination” of individual transmission lineages. The approach is able to overcome the noisy, obfuscated signals found in case incidence data. This work is generally of high quality. However, we feel that there is a need for additional effort to improve (1) the communication of their approaches and findings and (2) methodological rigor. In particular, the study implements an intricate methodology. The descriptions of analytical techniques and results are quite dense. The language could be simplified, and the jargon removed. Specific concerns are detailed below.

Major Concerns

- 1) The authors describe two key parameters from the Phydality algorithm, p and t , corresponding to the proportion p of all coalescent events in a phylogenetic tree occurring within a particular period t of the putative lineage’s root. These parameters are the basis from which individual transmission lineages are identified within larger phylogenetic trees. The authors describe how the identification of transmission lineages is sensitive to changes in the values of these parameters and state the values “were chosen to balance sensitivity and specificity” [of transmission lineage identification]. This a critical step in their analysis and the basis from which many inferences are made. Given the importance and centrality of these parameters to the analysis and conclusions, a more formal sensitivity analysis is warranted. In what ways does their sample composition change with changes to these parameter values? How do changes in the sample composition change the results?
- 2) Similarly, the authors use very specific values to estimate lineage establishment timing, i.e., they define lineage establishment timing as the first week the lineage had accounted for at least 5% of total incidence in that season. It is unclear how the authors chose the value of 5% to represent “substantial epidemic activity” and how this may impact their findings. This is particularly concerning as one of their main findings is stated as “Lineage size correlates with timing of establishment but not emergence (Line 132).” This is based on the magnitude of a correlation coefficient (Line 159). How might these analyses and conclusions change if different thresholds of total incidence are used? Do different thresholds affect the relative frequencies of influenza subtypes/lineages across transmission lineages? Is it possible to characterize establishment using metrics unrelated to the magnitude of incidence, e.g., lineage persistence? If so, how would this impact their study?
- 3) Could variation among influenza subtypes/lineages be an important addition to many steps in the study analyses? The dataset of genomic sequences used in the study is dominated by H3 (Line 547). While this is expected given different incidence rates and, potentially, virulence, this imbalance in the data could hide or obscure identified relationships for each subtype separately. How would the correlation analysis of lineage size and establishment timing differ if analyzed separately for each subtype? Are the conclusions valid for all influenza subtypes/lineages, or only the subtype dominating their sample? The authors acknowledge that differences among subtypes may be particularly important for the association analyses between viral spread and mobility, but also describe a limitation to characterize variation across subtypes/lineages (Lines 503-508). It is unclear why the comparisons cannot be made. While it may be difficult to determine the source of variation, it would be helpful to see how much variation exists between subtypes. Presenting this variation to the reader would help them to evaluate and interpret these results.

Minor Concerns

- Abstract language could be simplified and edited for readability

Ex. Lines 33&34 -

- “topology of human mobility yields repeatable patterns of which influenza viruses will circulate where”

Specifically, I'm not sure what topology of human mobility means, generally, this sentence is confusing. The paper would benefit from editing to simplify the writing and ensuring clarity.

- The method section is complicated. Conceptual figures, e.g., for cluster identification, would be helpful to ease reader comprehension. Why were certain techniques chosen over other options?

Ex. Lines 574-577

- “We clustered the taxa in each of these trees by computing the largest groups of viruses where, for each taxon within a cluster, there was at least one other taxon in the group that saw a patristic distance to the former taxon that was smaller than a given distance threshold.”

Relating state population size and sequencing rate, i.e., Lines 630-636, is there an advantage to this methodology over comparing with case counts?

Spread reconstruction, i.e., lines 673-696, is difficult to understand.

Line 643 fails to convey important details of smoothing methodology. Why is it necessary to smooth? Why is this smoothing technique used over others?

- “applied a 4253H, Twice smoother”

Calculation of type-specific incidence, e.g., lines 639-642, should cite previous works using this methodology and acknowledge limitations.

- E.g.. Amanda C Perofsky et al., 2024 eLife. <https://doi.org/10.7554/eLife.91849.1> (who refer to two other previous studies using this approach)

Why are HHS regions used in phylogeographic analyses, but Census regions are used for qualitative comparisons, i.e., lines 174-175 & Figure 2A?

- Also, in the descriptions of source regions, lines 227-232. HHS Regions and Census Regions group states in different ways, with some irreconcilable categorizations.

- Authors acknowledged the potential for reassortment events, i.e., line 588. Was there any evidence of reassortment identified? How might that have impacted cluster and/or transmission lineage identification?
- Additional discussion on alternative mobility mechanisms (e.g., local scale school commutes, leisure travel) or other resolutions not included in the study that might be important drivers of influenza spread.

- Typos

Line 629 *affecting

Line 646 missing comma after Latin abbreviation "i.e."; error throughout

Line 656 missing word after "particular"

- Figures

Figure 1A, what makes these phylogenies representative?

Supplementary figure 1 axis has single years. It is unclear to which season these refer.

(Remarks on code availability)

I reviewed that the scripts were available and that the code was well documented. I did not install or run the code

Reviewer #3

(Remarks to the Author)

This is a very nice paper and a welcome analysis of spatial diffusion of influenza across US states using sequence data. While several prior papers have focused on spatial diffusion using epidemiological time series, or on spatial diffusion using genetic data in other large countries (eg Australia), a phylogenetic analysis of US flu epidemics has for some reason eluded the community. Sequence analyses provide a more detailed understanding of spatial diffusion than epidemiological datasets. The paper is well written and presents interesting results that can be contrasted with earlier work.

I have a few broad comments on the paper:

1) The paper is focused on analysis of flu lineages circulating in the US, but it ignores introductions from outside the US. I would think that analysis of external seeding from outside of the US would contribute to a better understanding of seasonal spread, along with within country spread. It would be worth defining the scope of the study early in the introduction and returning to the potential role of external seeding in discussion.

2) How do the antigenic characteristics of the different lineages relate to the observed spatial patterns? Is it to be assumed that all lineages circulating in a given season have the same antigenic characteristics and more broadly, that they are equivalent in terms of viral fitness? Could antigenic characteristics/viral fitness explain some of the variability in establishment patterns and lineage size shown in Fig 1?

3) Authors use similarities in composition of lineages between states to study the spatial structure of epidemics (Fig 2). Is there evidence of variation in spatial structure between seasons and subtypes, especially in terms of the role of distance and radial diffusion? In epidemiological data, we find that diffusion is more radial/spatially driven in specific seasons, possibly linked to larger antigenic changes and more concentration of the disease among children. Relatedly, differences in the regional and global diffusion of flu B, AH1 and H3 have been reported in epidemiological and evolutionary datasets. Can the analyses in Fig 2 be broken down by season (alternatively, leave one season at a time?) and by subtype?

4) I enjoyed the analyses of commuting and air travel presented in Fig 4-5.

a. Is it possible to provide a metric of diffusion as a function of geographic distance, in addition to association with air travel and commuting (ie show a spatial transmission kernel)?

b. Across these analyses, it would be good to provide a formal comparison of models that consider commuting, air travel, and both. (eg an AIC like metric, or even correlation, of predicted vs observed onset times in each state, for each lineage)? While the commuter driven model looks better visually, it would be helpful to have a quantitative estimate (which could also be used to distinguish between more refined spatial models).

c. Does commuting explain the observed patterns better than simple distance?

5) A few more details on the metapopulation simulation model would be useful:

a) Define beta and gamma. How was beta set?

b) How was the model calibrated? Were movements scaled so that the timing and sequence of simulated epidemics matched observed ones?

c) Cross immunity between lineages is not explicitly accounted for in the equations provided.

6) Is there any relationship between the patterns described in this paper and the intensity and timing of flu epidemic each season?

7) It may be interesting to discuss how the US migration patterns evidenced here fit with what is known about the global circulation of flu viruses <https://www.science.org/doi/10.1126/science.adq3003> ... could also mention post pandemic period disruption, since a post pandemic season is included in the US analysis.

(Remarks on code availability)

Version 1:

Reviewer comments:

Reviewer #1

(Remarks to the Author)

All comments were sufficiently addressed in the rebuttal. No additional comments.

(Remarks on code availability)

Reviewer #2

(Remarks to the Author)

This is a thorough response to my concerns. I have no further comments at this time

(Remarks on code availability)

Reviewer #3

(Remarks to the Author)

The authors have carefully addressed all of the comments and added very useful clarification, sensitivity analyses, and extra caveats/discussion.

(Remarks on code availability)

Review of Jong et al. Nature Communications

In this article, de Jong and colleagues use a genomic centric approach to characterize Influenza outbreaks in the US. By combining influenza genomic data with a variety of other available data sources, including ILL-clinical data reported by the CDC and census-level commuter data, the authors were able to determine that influenza outbreaks are variable in size and lineage composition by year and region in the US, and lineage spread and competition is influenced by commuting patterns. The authors present a robust and well-analyzed dataset that tackles a standing question in influenza virus epidemiology: are epidemics in the US defined by single lineages, multiple co-circulating lineages, and does this vary from year to year and from state to state? In my opinion, the real challenge of a study like this is to develop a reasonable way to define transmission lineages, and I think the authors present a logical approach to defining influenza transmission lineages that is repeatable appropriately caveated. Collectively, this is an interesting manuscript that answers important questions using a robustly sampled dataset. I only have a few questions and points of clarification, listed below.

We thank the reviewer for their positive remarks and constructive comments.

1. Line 546: Have the authors determined how representative the sequencing dataset is in relation to the case burden caused by influenza sub-types? There are >30,000 genomes included in this study and I am curious if those are representative of the yearly incidence of influenza cases caused by each subtype, which I think is a straight forward question to answer. Given how much the authors determine transmission lineage and extrapolate lineage-specific incidence, it would be important to determine if some influenza infections are over represented in the sequencing data.

To address this question, we first computed the proportion A/H3N2 sequences among all influenza A sequences, and B/Yamagata sequences among all influenza B sequences, in each season in each state. We correlated these proportions with the proportion of A/H3N2 positive tests among flu A-positive tests and B/Yam-positive tests among flu B-positive tests in public health laboratories, respectively (orthogonal data that was not used elsewhere in the study). We found a strong correlation for flu A (Pearson $r = 0.79$) and flu B (Pearson $r = 0.83$). As such, we conclude that there is no reason to assume the sequence data are not representative of the subtypes/lineages' case burden. Furthermore, we note that our analyses of lineage size and lineage establishment timing are performed by individual subtype. For example, transmission lineage size is computed relative to all other transmission lineages of the same subtype, and the correlation of lineage size and lineage onset is computed only among lineages of the same subtype. Hence, even if there were a strong overrepresentation of some subtypes in the data, our methods are robust to this.

2. Line 580: The authors state each sequenced virus segment was assigned a segment-specific cluster identity. The authors then assigned a genome-wide cluster identity as the combination of individual segment identities. Did all the segment-specific identities need to be identical for viruses to be assigned the same genome-wide cluster identity? If not, how much variability was allowed? I am wondering if this approach would be able to account for potential recombination events, or if recombination would inherently result in a different genome-wide cluster identity.

Indeed, the aim of this procedure was to account for possible genome reassortment by ensuring that all viruses in a genome-wide cluster identity were genetically closely related for each segment, and building trees separately for each genome-wide cluster identity (which are then used to identify transmission lineages). In doing so, we minimized the risk of reassortment leading to errors in the tree construction. We have rephrased this section of the methods to make clearer that accounting for reassortment was the motivation behind this procedure. The segment-specific identities did not need to be identical for viruses to be grouped together. Rather, the pairwise distance between any viruses within a group was required to be smaller than the expected number of nucleotide substitutions in a two-year period. This variability was allowed because one would expect substantial genetic variability even within a genuine transmission lineage. We have rephrased to avoid such confusion by terming these groups 'segment-specific clusters' rather than 'segment-specific identities'.

"To minimize risk of bias in phylogenetic inference resulting from tree incongruence due to gene reassortment, we inferred whole-genome phylogenies separately for groups of viruses that were highly genetically related across all segments. To identify these groups of viruses, we constructed phylogenies for each segment individually using all viruses for the entire period using IQTree⁵² with a HKY⁵³ substitution model. We clustered these trees by identifying groups of viruses where, for each taxon within that group, there was at least one other taxon in the group that saw a patristic distance to the former taxon that was smaller than a given distance threshold. We defined this distance threshold as the expected number of mutations over a two-year period given the estimated molecular clock rate for that segment and subtype, using a more relaxed three-year period for the MP and NS segments for additional lenience given their lower evolutionary rate. Using these segment-specific clusters, we assigned each taxon a genome-wide cluster as the combination of the taxon's segment-specific clusters." (lines 616-628)

3. Line 642: Did estimates of lineage-specific incidence match the clinical ILL type-specific incidence from year to year? I think this comparison would also be a way to determine if some influenza types are over- or under-represented in the genomics dataset compared to true incidence caused by each influenza subtype.

In our analyses, we estimated lineage-specific incidence from the type-specific ILI signal. See response to comment 1 by this reviewer above.

4. Line 656: I believe a word is missing in this sentence.

Fixed. We thank the reviewer for noting the omission.

5. Using this dataset, would it be feasible to assess the year-to-year composition of transmission lineages to specifically ask the question if any lineages are maintained between seasons? While this seems unlikely due to influenza seasonality, but I wonder the same transmission lineages are ever identified in multiple seasons, presumably because it has maintained transmission globally.

This question is more related to the global dynamics of virus persistence and spread (e.g. Bedford et al, Nature 2015), rather than the within-US dynamics the paper is focused on. If a virus were to circulate in between seasons and be re-introduced in the subsequent season, it is unlikely going to be detected as a single transmission lineage, which is identified as a large number of successive coalescent events occurring early during the subsequent season. It is more likely that each reintroduction would lead to a separate rapid succession of coalescent event during the season if that were the case. While it is possible for a transmission lineage to seed transmission chains abroad and be re-introduced the next season, the probability for any given lineage to do so is vanishingly small given influenza viruses' large global diversity. This could nevertheless happen and the United States is known to have played a role as a source for influenza's global metapopulation dynamics in some seasons. We note the US' role in the global circulation network in the Introduction: "*Analyses of sequence data have shown that a highly dynamic global viral metapopulation, driven by complex patterns of inter-regional viral migration, re-seeds US epidemics each year²⁷. In turn, these epidemics can go on to seed epidemics elsewhere^{13,27}. However, the processes that turn initial epidemic seeds in the US, introduced from abroad, into full-fledged epidemics remain poorly understood.*" (lines 48-52). We note that we do find a plausible example of season-to-season local persistence in the 2019/2020 B/Vic season in Florida –this example is supported by genetic sampling of that transmission lineage throughout the summer and supported by clinical detection of influenza B viruses in Florida throughout the summer.

Reviewer #2 (Remarks to the Author):

The authors sought to characterize transmission in outbreaks of seasonal influenza within the United States. It is challenging to study seasonal influenza dynamics as outbreaks are often comprised of many concurrent chains of transmission, especially from the perspective of larger spatial scales. The authors utilize genomic epidemiology and mathematical modeling techniques to characterize "epidemic emergence, establishment and viral

dissemination” of individual transmission lineages. The approach is able to overcome the noisy, obfuscated signals found in case incidence data. This work is generally of high quality. However, we feel that there is a need for additional effort to improve (1) the communication of their approaches and findings and (2) methodological rigor. In particular, the study implements an intricate methodology. The descriptions of analytical techniques and results are quite dense. The language could be simplified, and the jargon removed. Specific concerns are detailed below.

We thank the reviewer for their positive remarks and constructive comments.

Major Concerns

1) The authors describe two key parameters from the Phydely algorithm, p and t , corresponding to the proportion p of all coalescent events in a phylogenetic tree occurring within a particular period t of the putative lineage’s root. These parameters are the basis from which individual transmission lineages are identified within larger phylogenetic trees. The authors describe how the identification of transmission lineages is sensitive to changes in the values of these parameters and state the values “were chosen to balance sensitivity and specificity” [of transmission lineage identification]. This a critical step in their analysis and the basis from which many inferences are made. Given the importance and centrality of these parameters to the analysis and conclusions, a more formal sensitivity analysis is warranted. In what ways does their sample composition change with changes to these parameter values? How do changes in the sample composition change the results?

We appreciate the reviewer’s point about the sensitivity of our conclusions to the parameters used to delineate the sequences into transmission lineages. We performed a sensitivity analysis by keeping the period t constant (at 1 month to account for week-to-week variation in sequencing numbers), and varying p so it took on values 0.05, 0.10, 0.15, and 0.20. Supplementary Fig. 6 shows the analogous results presented in Fig. 1b-e for different values of p . For very small p , a smaller number of clusters is identified (as the clustering procedure is less stringent, and hence larger groups of viruses can be subsumed into a single cluster), whereas when p is high, the number of clusters is greater (as larger clusters found for smaller p need to be broken down into multiple subtrees that individually do satisfy the strict requirement). While this variation exists, the sensitivity analyses shown in Supplementary Fig. 6 indicate that our conclusions are robust to the choice of p . In analyses for individual seasons (Fig. 3 & 4), we have explicitly chosen to include the phylogenies themselves to allow the reader to judge that the identified clusters are reasonable (with the identified spatial signal for individual lineage lending credence to the validity of the clustering method).

2) Similarly, the authors use very specific values to estimate lineage establishment timing, i.e., they define lineage establishment timing as the first week the lineage had accounted for

at least 5% of total incidence in that season. It is unclear how the authors chose the value of 5% to represent “substantial epidemic activity” and how this may impact their findings. This is particularly concerning as one of their main findings is stated as “Lineage size correlates with timing of establishment but not emergence (Line 132).” This is based on the magnitude of a correlation coefficient (Line 159). How might these analyses and conclusions change if different thresholds of total incidence are used? Do different thresholds affect the relative frequencies of influenza subtypes/lineages across transmission lineages? Is it possible to characterize establishment using metrics unrelated to the magnitude of incidence, e.g., lineage persistence? If so, how would this impact their study?

We repeated our analyses with different incidence thresholds (0.1%, 0.5%, 1%, 2%, 5%, 10%). The correlation analyses using these various thresholds are visualized in Supplementary Fig. 7. In these analyses, we see that as the incidence threshold gets smaller, there are more ‘early-establishing’ lineages that do not actually cause substantial epidemic activity in the corresponding season (particularly when the incidence threshold <2%). This shows precisely the point of the analysis: only when a lineage actually causes substantial levels of epidemic activity, can one say with relative confidence that it is likely to be successful. In contrast, simply sampling the virus, without evidence that it is substantially propagating through the population, does not give substantial information about whether a lineage will be successful or not (as shown in Fig. 1d). We have rephrased the section to illustrate this point:

“However, we hypothesized that if a lineage did cause substantial epidemic activity early on, it would likely be successful nationwide. We defined the timing of lineage establishment as the first week in which the lineage accounted for substantial epidemic activity in at least one state. When defining lineage establishment week as the first week a lineage had cumulatively accounted for >5% of the state’s total epidemic activity of that subtype, establishment week correlated strongly with nationwide lineage size for all subtypes (Spearman $\rho = -0.46, -0.50, -0.70, -0.71$ for A/H3N2, A/H1N1pdm09, B/Yam, B/Vic, respectively, $P < 0.001$ for all) (Fig. 1e). Statistical modelling indicated each week’s delay in establishment resulted in a decrease in lineage size of 16.9%, 15.1%, 14.6%, and 13.0% for A/H3N2, A/H1N1pdm09, B/Yam, and B/Vic, respectively ($P < 0.001$ for all). The fact that the lineages that first established substantial epidemic activity somewhere in the US were more likely to be successful suggests that the states with the earliest epidemic onset have outsized contributions to which viruses will circulate nationwide. The proportion of the earliest-establishing transmission chains that resulted in a successful transmission lineage decreased as the cumulative incidence threshold used to determine establishment week decreased (Supplementary Fig. 7). This suggests that while a first-mover advantage shapes lineage success, substantial levels of epidemic activity are necessary for a lineage’s early circulation to be predictive of substantial nationwide spread.” (lines 147-164).

3) Could variation among influenza subtypes/lineages be an important addition to many steps in the study analyses? The dataset of genomic sequences used in the study is dominated by H3 (Line 547). While this is expected given different incidence rates and, potentially, virulence, this imbalance in the data could hide or obscure identified relationships for each subtype separately. How would the correlation analysis of lineage size and establishment timing differ if analyzed separately for each subtype? Are the conclusions valid for all influenza subtypes/lineages, or only the subtype dominating their sample?

We refer the reviewer to our response to reviewer 1 regarding the representativeness of the genomic data. Our correlation analyses of size and timing are performed separately for each subtype in Fig. 1e. However, in our initial submission, we did not report the results by subtype in the main text. In our revised submission, we report summary statistics and p-values for each subtype. To improve rigor in this analysis, we also performed a mixed-effect regression analysis that estimates the effect of time on size, incorporating season-to-season variation, in addition to the existing correlation analyses. This analysis was performed for each season individually to ensure the outcomes were robust to the biases suggested by the reviewer:

When defining lineage establishment week as the first week a lineage had cumulatively accounted for >5% of the state's total epidemic activity of that subtype, establishment week correlated strongly with nationwide lineage size for all subtypes (Spearman $\rho = -0.46, -0.50, -0.70, -0.71$ for A/H3N2, A/H1N1pdm09, B/Yam, B/Vic, respectively, $P < 0.001$ for all) (Fig. 1e). Statistical modelling indicated each week's delay in establishment resulted in a decrease in lineage size of 16.9%, 15.1%, 14.6%, and 13.0% for A/H3N2, A/H1N1pdm09, B/Yam, and B/Vic, respectively ($P < 0.001$ for all). (lines 150-156).

The authors acknowledge that differences among subtypes may be particularly important for the association analyses between viral spread and mobility, but also describe a limitation to characterize variation across subtypes/lineages (Lines 503-508). It is unclear why the comparisons cannot be made. While it may be difficult to determine the source of variation, it would be helpful to see how much variation exists between subtypes. Presenting this variation to the reader would help them to evaluate and interpret these results.

We appreciate the reviewer's point. In Supplementary Fig. 8, we now present MDS analyses showing the spatial structure of lineage compositions by season and subtype. In Supplementary Fig. 9, we now show the correlation between compositional similarity and pairwise distance by season and subtype. In Supplementary Fig. 15 we now show the correlation between pairwise distance and the normalized pairwise viral migration frequency as estimated from phylogeographic analyses. We have also added further detail in the discussion about the caveats with regard to comparison between subtypes and seasons:

“Our analyses sought to identify differences among subtypes and seasons in their dynamics of viral spread. Mobility flows underlying the spread of influenza B viruses are potentially different from those for influenza A viruses as a result of differences in the age distribution of infection²⁷. Furthermore, viral spread has been suggested to be particularly localized in seasons that saw the circulation of a novel antigenic variant³⁰. While we identified differences among individual seasons and subtypes in their spatial structure and the distance-dependence of viral spread, we are cautious about direct comparison across seasons and subtypes. For example, the lower evolutionary rate of influenza B viruses could result in reduced detectable spatial signal, as the increased time between successive mutations would lead to an attenuated evolutionary imprint of the viral migration processes. If a lineage spreads particularly quickly, the spatial signal could be similarly obscured. While our analyses are unable to confidently ascertain differences among subtypes, our mechanistic simulations were able to recapitulate observed patterns of spread using commuting data for influenza A and B viruses, suggesting broadly similar mechanisms drive the spread of both.” (lines 533-547).

Minor Concerns

- Abstract language could be simplified and edited for readability

Ex. Lines 33&34 -

- “topology of human mobility yields repeatable patterns of which influenza viruses will circulate where”

Specifically, I'm not sure what topology of human mobility means, generally, this sentence is confusing. The paper would benefit from editing to simplify the writing and ensuring clarity.

We have edited the abstract to improve clarity and ensure compliance with *Nature Communications* guidelines. We have edited the paper to maximize clarity and accessibility.

- The method section is complicated. Conceptual figures, e.g., for cluster identification, would be helpful to ease reader comprehension. Why were certain techniques chosen over other options?

Ex. Lines 574-577

- “We clustered the taxa in each of these trees by computing the largest groups of viruses where, for each taxon within a cluster, there was at least one other taxon in the group that saw a patristic distance to the former taxon that was smaller than a given distance threshold.”

We have made clearer the rationale behind this process: to account for the confounding effect of reassortment. We agree that this was insufficiently clear in the initial submission (see response to reviewer 1).

Relating state population size and sequencing rate, i.e., Lines 630-636, is there an advantage to this methodology over comparing with case counts?

We agree that directly comparing with case counts rather than population size would make intuitive sense. However, one runs into the issue of how representative case counts are of actual incidence. The CDC also cautions against the use of case counts (“The number of specimens tested and % positive rate vary by region and season based on different testing practices including triaging of specimens by the reporting labs, therefore it is not appropriate to compare the magnitude of positivity rates or the number of positive specimens between regions or seasons.”

(<https://gis.cdc.gov/grasp/fluview/fluportaldashboard.html>)). The strong relationship identified between population size and sequencing rate suggests there is an underlying relationship between the two quantities.

Spread reconstruction, i.e., lines 673-696, is difficult to understand.

We have simplified the section about reconstructions of spatial spread:

“To extract transmission lineage-specific epidemic curves, we fitted the sampling dates of taxa belonging to individual transmission lineages to the reconstructed ILI+ curves. Specifically, for each state and type (i.e., influenza A or B) individually, we computed the contribution of each taxon in each week using a normal density with mean equal to the taxon’s sampling date, with a two-week standard deviation. In each week, the contribution of a transmission lineage to epidemic activity in a state was given by the sum of taxon contributions in that week from viruses belonging transmission lineage in question, divided by the sum of taxon contributions in that week across all viruses. The weekly relative incidence of a transmission lineage was then computed by multiplying the transmission lineage’s contribution in that week by that week’s ILI+ signal for the corresponding type. We retained taxa that were not assigned to any transmission lineage as a separate group (‘unclustered’).” (lines 669-680)

Line 643 fails to convey important details of smoothing methodology. Why is it necessary to smooth? Why is this smoothing technique used over others?

- “applied a 4253H, Twice smoother”

We have added more detail about the rationale and choice of smoothing procedure:

‘Following previous work⁵⁷, we applied a 4253H, Twice smoother, implemented in the sleekts R package, to smooth the epidemic curves, in order to increase robustness of estimation of lineage establishment weeks.’ (lines 667-669).

Calculation of type-specific incidence, e.g., lines 639-642, should cite previous works using this methodology and acknowledge limitations.

- E.g., Amanda C Perofsky et al., 2024 eLife. <https://doi.org/10.7554/eLife.91849.1> (who refer to two other previous studies using this approach)

We have added appropriate citations.

Why are HHS regions used in phylogeographic analyses, but Census regions are used for qualitative comparisons, i.e., lines 174-175 & Figure 2A?

- Also, in the descriptions of source regions, lines 227-232. HHS Regions and Census Regions group states in different ways, with some irreconcilable categorizations.

In phylogeographic analyses, the choice of spatial scale is essential to the analysis; a scale too granular could impede the detection of spatial signal if sample sizes are too low for the individual geographical groupings, while geographical groupings that are too large could obscure salient features of the data within those groupings. In our phylogeographic analyses, HHS regions represent the best-performing spatial grouping: sample sizes at the level of individual states would be too low for robust analyses, while analyses at the Census region level would obscure potentially interesting patterns within different spatial groupings *within* the Census region. HHS regions have sufficient overall samples for robust phylogeographic analyses while retaining the maximum granularity possible.

We understand that this leaves potentially awkward irreconcilable differences between the regions. In particular, the way the source-sink analyses were presented (e.g., “three likely first established in the South (HHS regions 4 & 6)”) results in confusion because the South is a designated Census Region whereas here it refers to the broader region encompassed by the HHS regions 4 and 6. We have rephrased to make clear that these represent broader geographic regions rather than specific Census regions (e.g., “three likely first established in the southern United States (HHS regions 4 & 6)”).

We opted for Census regions for the qualitative comparisons as these allow for easier visualization (with ten colors instead of four, Figs 2a and 2b would not be as easily interpretable). Furthermore, an advantage to Census regions is that they are more easily understood (e.g., South is a more intuitive notion to a reader than HHS region 4) and this helps take the reader through the narrative.

- Authors acknowledged the potential for reassortment events, i.e., line 588. Was there any evidence of reassortment identified? How might that have impacted cluster and/or transmission lineage identification?

To minimize the potential for reassortment biasing the tree reconstructions, we generated whole-genome trees for groups of viruses that were genetically highly related across all segments. These are the trees we use as input for the procedure used to detect transmission lineages. It is likely that there would have been reassortment at some point in viruses' shared evolutionary history – we do not explicitly aim to detect reassortment events, but rather ensure that they do not affect the validity of the results by leading to large biases in tree reconstruction. If reassortment happened among viruses that were highly similar, it could be that this was not accounted for using the above procedure. However, this would not substantially highly affect our results, as they would simply not be clustered in the procedure used to detect transmission lineages, lying on different branches in the whole-genome phylogenies.

- Additional discussion on alternative mobility mechanisms (e.g., local scale school commutes, leisure travel) or other resolutions not included in the study that might be important drivers of influenza spread.

We have added further discussion of the potential roles for alternative mobility mechanisms in the Discussion:

“Our metapopulation analyses provide a framework to fit models of human mobility to epidemic pathogens in the presence of lineage structure, which could be used to evaluate more refined spatial models and their parameterizations in future work. However, it is striking that we could reproduce lineage spread dynamics using mechanistic simulations when parameterizing mobility directly using commuter surveys, without fitting any mobility-related parameters. This indicates that commuting flows provide a parsimonious explanation for observed patterns of viral spread. We could only perform our analyses at the state level owing to that being the level of spatial resolution in most virus metadata, and analyses at other spatial scales may yield different results regarding modes of virus spread^{30,42,43}. Furthermore, our results do not rule out potential roles for alternative types of mobility, such as movement of children not well-captured by commuting flows⁴⁴.” (lines 507-517)

- Typos

Line 629 *affecting

Line 646 missing comma after Latin abbreviation “i.e.”; error throughout

Line 656 missing word after “particular”

We thank the reviewer for the corrections.

- Figures

Figure 1A, what makes these phylogenies representative?

Supplementary figure 1 axis has single years. It is unclear to which season these refer.

We have removed the term 'representative' in figure 1a. We have fixed the labels to refer to the exact season.

Reviewer #2 (Remarks on code availability):

I reviewed that the scripts were available and that the code was well documented. I did not install or run the code

Reviewer #3 (Remarks to the Author):

This is a very nice paper and a welcome analysis of spatial diffusion of influenza across US states using sequence data. While several prior papers have focused on spatial diffusion using epidemiological time series, or on spatial diffusion using genetic data in other large countries (eg Australia), a phylogenetic analysis of US flu epidemics has for some reason eluded the community. Sequence analyses provide a more detailed understanding of spatial diffusion than epidemiological datasets. The paper is well written and presents interesting results that can be contrasted with earlier work.

We thank the reviewer for their positive remarks and constructive comments.

I have a few broad comments on the paper:

1) The paper is focused on analysis of flu lineages circulating in the US, but it ignores introductions from outside the US. I would think that analysis of external seeding from outside of the US would contribute to a better understanding of seasonal spread, along with within country spread. It would be worth defining the scope of the study early in the introduction and returning to the potential role of external seeding in discussion.

We have amended the introduction to make clearer that our analyses are focused on the processes of epidemic establishment following external seeding into the United States:

“Analyses of sequence data have shown that a highly dynamic global viral metapopulation, driven by complex patterns of inter-regional viral migration, re-seeds US epidemics each year²⁷. In turn, these epidemics can go on to seed epidemics elsewhere^{13,27}. However, the processes that turn initial epidemic seeds in the US, introduced from abroad, into full-fledged epidemics remain poorly understood.” (lines 48-52).

We have also added broader discussion of the role of external seeding in the Discussion:

“Our analyses reveal the frequent early establishment and national success of lineages emerging in the South. However, this pattern was not consistent across seasons, and our results demonstrate a striking diversity and season-to-season variability in the location of emergence of successful transmission lineages. A key question is what drives this variability. Potential explanations include year-to-year differences among regions in environmental factors that shape early-season transmission potential³⁹, or differences among regions in susceptibility to circulating viruses, possibly related to patterns of viral circulation in prior seasons^{2,40}. Alternatively, differences among states in the supply of epidemic seeds from abroad could lead to differences in the locations of initial lineage establishment. Such effects could arise from among-state differences in the preferred destinations for travelers¹², in combination with year-to-year variability in where substantial circulation of viruses with high fitness occurs globally^{13,14,27,41} in the period when epidemic seeds would find fertile ground for transmission in the US. Testing this hypothesis and leveraging the predictive value that may be gained for prediction of epidemic composition underscores the need for strong global genomic and epidemiological surveillance networks.” (lines 484-498).

2) How do the antigenic characteristics of the different lineages relate to the observed spatial patterns? Is it to be assumed that all lineages circulating in a given season have the same antigenic characteristics and more broadly, that they are equivalent in terms of viral fitness? Could antigenic characteristics/viral fitness explain some of the variability in establishment patterns and lineage size shown in Fig 1?

We appreciate the reviewer’s point about the role of viral fitness in lineage dynamics. Our initial manuscript alluded to the role of viral fitness in shaping season-to-season variability in spatial patterns, such as the rapid expansion of the dominant A/H3N2 lineage in the 2018/2019 season in which a new antigenic variant circulated. However, we agree that a more comprehensive discussion of the factors that shape the fitness of any given lineage would enrich the manuscript. We have added discussion regarding the role of antigenic characteristics in shaping lineage success:

“Our results indicate that competition for susceptible individuals is a primary determinant of lineage success: lineages that establish earlier and in better-connected states are on average larger. Our simulations assume all individuals are equally susceptible to all lineages, but differences in antigenic properties among lineages could potentially explain further variation in lineage success. For example, US influenza seasons where circulating viruses saw more mutations in antigenic sites of the hemagglutinin (HA) and neuraminidase (NA) surface proteins were associated with larger epidemics⁴⁰, and the viral composition of the 2017/2018 US A/H3N2 season was consistent with dominance of the viruses to which population immunity as measured using standard serological assays was lowest⁴⁵. This hypothesis is also consistent with, for example, the dominance of the antigenically drifted lineage in the 2018/2019 A/H3N2 season³⁶. However, our study is underpowered to detect

fitness differences between lineages that could be attributed to antigenic change, due to the study's relatively noisy data and limited time period, confounding due to orthogonal determinants of lineage fitness (such as environmental conditions in the state of emergence), combined with the complexity of influenza virus antigenic dynamics^{4,6}.” (lines 549-563)

A vital observation in our study is that we saw transmission lineages sweeping across the US that were highly genetically related to viruses that had already been circulating. This indicates that antigenic characteristics do not fully explain differences in lineage success. We make this point in the Discussion:

“An essential question is what drives the heterogeneity in the speed of establishment of lineages. We found that many of the most successful transmission lineages emerged very shortly before nation-wide epidemic onset and established rapidly, sometimes sweeping to national dominance despite substantial competition from other contemporaneous transmission chains. In some seasons the outcompeted contemporaneous lineages were highly genetically related to the lineage that would come to dominate, with a common ancestor in or after the summer preceding the corresponding season. Importantly, this suggests that viral factors such as antigenic novelty are unable to fully explain the heterogeneity in the speed of lineage establishment.” (lines 565-573)

3) Authors use similarities in composition of lineages between states to study the spatial structure of epidemics (Fig 2). Is there evidence of variation in spatial structure between seasons and subtypes, especially in terms of the role of distance and radial diffusion? In epidemiological data, we find that diffusion is more radial/spatially driven in specific seasons, possibly linked to larger antigenic changes and more concentration of the disease among children. Relatedly, differences in the regional and global diffusion of flu B, AH1 and H3 have been reported in epidemiological and evolutionary datasets. Can the analyses in Fig 2 be broken down by season (alternatively, leave one season at a time?) and by subtype?

We have broken down the analyses of transmission lineage spatial structure (Supplementary Fig. 8 & 9) and the analyses linking phylogeographic reconstructions of spread to mobility (Supplementary Fig. 15) by subtype and season.

We find differences among seasons and subtypes in the degree to which spread is spatially driven. For example, in the correlation between pairwise viral jump frequency and distance (Supplementary Figure 15) - intriguingly, we find that of the three seasons with the strongest correlation (2016/2017 A/H3N2, 2018/2019 A/H1N1pdm09, and 2022/2023 A/H3N2), the first two did not see the circulation of antigenically novel viruses. Hence, our results do not support the notion that antigenic changes lead to more spatially driven epidemics. Furthermore, the fact that these three were all influenza A viruses runs contrary to the notion that flu B viruses should exhibit more spatial structure. However, we believe that

these differences should not be overinterpreted. For example, a lower mutation rate (such as seen for flu B viruses), particularly at the short timescales of a single season, would *a priori* lead to less spatial structure. Similarly, if sampling rates are lower, less spatial structure would be detectable. Finally, if a season saw more co-circulating lineages, this would likely lead to a stronger correlation with distance, as competition constrains spread (this is for example what we see in the 2016/2017 A/H3N2 and 2018/2019 A/H1N1pdm09 seasons). Because all these different variables are difficult to disentangle, we refrain from making broad inferences based on individual seasons even though we have presented the estimates for individual seasons in the Supplementary Figures. We note the difficulties in interpreting differences among seasons and subtypes in the Discussion:

“Our analyses sought to identify differences among subtypes and seasons in their dynamics of viral spread. Mobility flows underlying the spread of influenza B viruses are potentially different from those for influenza A viruses as a result of differences in the age distribution of infection²⁷. Furthermore, viral spread has been suggested to be particularly localized in seasons that saw the circulation of a novel antigenic variant³⁰. While we identified differences among individual seasons and subtypes in their spatial structure and the distance-dependence of viral spread, we are cautious about direct comparison across seasons and subtypes. For example, the lower evolutionary rate of influenza B viruses could result in reduced detectable spatial signal, as the increased time between successive mutations would lead to an attenuated evolutionary imprint of the viral migration processes. If a lineage spreads particularly quickly, the spatial signal could be similarly obscured. While our analyses are unable to confidently ascertain differences among subtypes, our mechanistic simulations were able to recapitulate observed patterns of spread using commuting data for influenza A and B viruses, suggesting broadly similar mechanisms drive the spread of both.” (lines 533-547)

4) I enjoyed the analyses of commuting and air travel presented in Fig 4-5.

a. Is it possible to provide a metric of diffusion as a function of geographic distance, in addition to association with air travel and commuting (ie show a spatial transmission kernel)?

In addition to our analyses which explicitly use air travel and commuting data, we have added a more general singly-constrained gravity model where the propensity p_{ij} for an individual from state j to travel to state i (conditional on travelling) depends on state i 's population size N_i and the distance between states d_{ij} as $p_{ij} \propto N_i^\tau / d_{ij}^\rho$. Here, we informed p_{ij} using a combination of states' commuting and air travel rates. Using this model, we estimate the decay of diffusion with distance:

“To derive a more general quantification of the gravity-like nature of viral spread, we fit a model where the propensity p_{ij} for an individual from state j to travel to state i (conditional on travelling) depends on state i 's population size N_i and the distance between states d_{ij} as $p_{ij} \propto N_i^\tau / d_{ij}^\rho$. In this model, the daily probability for an individual from a given state to travel was informed by the commuting and air travel data. This general model accurately recapitulated observed lineage dynamics, with a strong fit to sample counts ($\ell = -329.7$), but with moderately decreased fit to establishment timing ($MSE = 9.5$) (Supplementary Fig. 13). We estimated a rapid decay with distance ($\rho = 1.4$, 95% CI 1.2 - 1.5) with a statistically significant dependence on destination population size ($\tau = 0.5$, 95% CI 0.3 - 0.8) (Supplementary Fig. 14).” (lines 395-404).

We note that if we do not use state-specific propensities to travel (informed by states' commuting and air travel data), but rather use a constant outward travel rate across all states (equal to the median of states' outward travel rates estimated from commuting and air travel data), the fit is worse than any of the models directly informed by commuting or air travel data. Hence, incorporating information about state-specific differences in outward travel rates is necessary for model fit, which provides evidence against the notion that simple distance would yield an equivalent fit (reviewer's point c below):

“However, if we assumed the daily probability of traveling was constant across states, the model yielded a substantially worse fit than the models directly informed by commuting and air travel data ($\ell = -429.6$, $MSE = 11.2$). This indicates that accounting for differences among states in outward travel rates is necessary to capture salient features of influenza virus spread.” (lines 404-408).

We performed the gravity model analyses only for the 2019/2020 B/Vic season, as the large number of lineages was necessary to render the model identifiable. For example, in the 2018/2019 season, with only two major lineages, perturbing the onset dates yielded substantially different parameter estimates (a particular lineage spreading less widely could be the result of a later onset but could also be the result of a rapid decay with distance). In the 2019/2020 B/Vic season, the constraints imposed by the larger number of seasons meant that the gravity models were robust to perturbations in the onset timings. This gravity fit serves as a proof-of-principle for fitting more complex models. However, the fact that we could reproduce observed spread without any fitted parameters using commuting data across multiple seasons and subtypes suggests that commuting yields a parsimonious explanation of observed patterns of spread, and this is the framing we use throughout. We note the point that our models provide a framework for the estimation of more detailed models and that our results do not necessarily rule out alternative models in the Discussion:

“Our results suggest that viral migration is well-described by commuting flows, which generate the network on which co-circulating lineages compete for disease-susceptible

individuals. Commuting data has previously been suggested to drive influenza viral spread based on analyses of ILI data^{12,30}, but this has not been shown mechanistically or validated against phylogenetically supported instances of viral spread across (sub)types^{11,19,30}. While we found a clear dominance of commuting over air travel when considering these metrics in isolation, our results also suggest that air travel flows not captured in commuter surveys could play a role in viral dissemination. Our metapopulation analyses provide a framework to fit models of human mobility to epidemic pathogens in the presence of lineage structure, which could be used to evaluate more refined spatial models and their parameterizations in future work. However, it is striking that we could reproduce lineage spread dynamics using mechanistic simulations when parameterizing mobility directly using commuter surveys, without fitting any mobility-related parameters. This indicates that commuting flows provide a parsimonious explanation for observed patterns of viral spread. We could only perform our analyses at the state level owing to that being the level of spatial resolution in most virus metadata, and analyses at other spatial scales may yield different results regarding modes of virus spread^{30,42,43}. Furthermore, our results do not rule out potential roles for alternative types of mobility, such as movement of children not well-captured by commuting flows⁴⁴.” (lines 500-517).

b. Across these analyses, it would be good to provide a formal comparison of models that consider commuting, air travel, and both. (eg an AIC like metric, or even correlation, of predicted vs observed onset times in each state, for each lineage)? While the commuter driven model looks better visually, it would be helpful to have a quantitative estimate (which could also be used to distinguish between more refined spatial models).

We understand the reviewer’s point and have made the model comparison more quantitative. We now explicitly quantify model fit by computing the mean squared error of the predicted vs. simulated lineage onset timings. In addition, we use a multinomial likelihood for the observed sequence counts in each state across the lineages included in the simulations, given the simulated proportion of infections attributable to each simulated lineage. This likelihood quantifies the degree to which the simulated distribution of lineages across states (as generated by mobility-induced competition) fits to the observed lineage sample counts in each state. We compare models by comparing the timing MSE and the sample count log-likelihood.

c. Does commuting explain the observed patterns better than simple distance?

As discussed in (a), we find evidence in the mechanistic simulations that commuting explains observed patterns better than simple distance. We also added an explicit correlation of centroid distance and pairwise migration rate as estimated in the phylogeographic analyses. Here, too, we find that commuting outperforms simple distance:

“We found that the states with the greatest jump contribution to another state tended to be the states that were most strongly connected through commuting flows to that state (Spearman $\rho = 0.63$, $P < 0.001$) (Fig. 5a). The correlation with air travel was weaker (Spearman $\rho = 0.32$, $P < 0.001$) (Fig. 5b), as was the correlation with state centroid distance (Spearman $\rho = -0.39$, $P < 0.001$). This provides further support that commuting is a strong correlate of the mobility processes that disseminate seasonal influenza viruses across the United States.” (lines 425-431).

5) A few more details on the metapopulation simulation model would be useful:

a) Define beta and gamma. How was beta set?

b) How was the model calibrated? Were movements scaled so that the timing and sequence of simulated epidemics matched observed ones?

We have added more detail regarding the model details and calibration in the Methods:

“ S_{ij} represents the number of susceptible individuals originating from state j that are currently in state i and I_{vij} represents the number of individuals originating from state j that are currently in state i and are infected by a virus corresponding to transmission lineage v . l_{ij} represents the outward travel rate from j to i and r represents the return rate. We assumed $r = 1 \text{ d}^{-1}$, recovery rate $\gamma = 4 \text{ d}^{-1}$, and $R_0 = 1.35^{12}$, with transmission rate $\beta = R_0 \times \gamma$.” (lines 789-793)

For the commuting analyses, we computed the number of commuters between each pair of states by aggregating across origin and destination counties in each state. We symmetrized these counts (though they are highly symmetric, $r = 0.9998$) and computed the daily travel rate l_{ij} as the number of symmetrized commuters between states i and j divided by the population size of state j . For air travel analyses, we computed the rate of air travel between states as the symmetrized total number of passengers in 2016 between airports located in the two states. We computed the daily travel rate l_{ij} as the yearly trip count divided by 365 and the population size of state j . For the simulations that used a combination of air travel and commuting flows, l_{ij} was defined the maximum of the pairwise air travel and commuting rates to account for the possibility that some of the commuting flows were accounted for by the air travel data.” (lines 795-805).

“To simulate competitive spread of lineages, we initialized each lineage in its first week of establishment in its likely onset state and simulated forward in time. We used the reconstructed first week of establishment in any state, rather than in the onset state, to account for situations where incidence data was absent for the likely onset state. In the simulations for each mobility modality, we allowed each lineage’s onset week to vary to up to two weeks after or two weeks before its estimated onset week, to account for error in the estimation of establishment timing with noisy data and situations where no incidence data

was available for the putative index state. For each individual mobility modality, we retained the configuration with the highest likelihood. Lineages were initialized with an infected population of 1×10^{-5} times the index state's population size. Analogous to the ground truth reconstructions, we computed the size of each lineage in the reconstructions as the proportion of infections across simulated lineages in a state that was attributable to a particular lineage and the week of establishment as the first week a lineage had caused >5% of total infections across simulated lineages. We computed the likelihood (ℓ) for each simulation as the product across all states of the multinomial probability of the observed sample counts for the transmission lineages given the relative proportions of infections attributable to the respective lineages in the simulations. We further quantified model fit with the mean squared error (MSE) of simulated vs. estimated lineage establishment timings. To fit the gravity model, we assumed the total daily rate of movement from each state was either 1) given by the sum of the daily outflows computed from the commuting data and the air travel data for that state, or 2) constant across states, given by the median of the state-specific rates computed in (1). We assumed the best-fit lineage onset weeks as estimated for the simulations using a combination of commuting and air travel. Given the total rate of movement from state j , we computed the flux to state i as $p_{ij} \propto N_i^\tau / d_{ij}^\rho$. We simulated for ρ from -0.5 to 3 and τ from -0.5 to 2, each in increments of 0.1. We computed the 95% CI based on profile likelihoods given the simulations for each combination of ρ and τ ." (lines 810-834)

c) Cross immunity between lineages is not explicitly accounted for in the equations provided.

We thank the reviewer for noting the omission. We have amended the equations to reflect cross-immunity between co-circulating lineages.

6) Is there any relationship between the patterns described in this paper and the intensity and timing of flu epidemic each season?

To address this question, we performed a correlation analysis between season intensity (as computed by e.g. Perofsky et al. <https://doi.org/10.7554/elife.91849.2>) and lineage diversity (quantified as the season-level Shannon entropy of lineage composition). We performed the same analysis for lineage diversity and season-specific epidemic onset. However, we found no consistent effect: "We found no relationship between transmission lineage diversity and metrics of epidemic timing (Pearson $r = -0.16$, $P = 0.63$) or intensity (Pearson $r = -0.12$, $P = 0.73$)." (lines 116-118).

7) It may be interesting to discuss how the US migration patterns evidenced here fit with what is known about the global circulation of flu

viruses <https://www.science.org/doi/10.1126/science.adq3003> ... could also mention post pandemic period disruption, since a post pandemic season is included in the US analysis.

We now more explicitly mention the role of the US in the global circulation of flu:

“Analyses of sequence data have shown that a highly dynamic global viral metapopulation, driven by complex patterns of inter-regional viral migration, re-seeds US epidemics each year²⁷. In turn, these epidemics can go on to seed epidemics elsewhere^{13,27}.” (lines 48-50)

and how global migration patterns might shape dynamics in the United States:

“Alternatively, differences among states in the supply of epidemic seeds from abroad could lead to differences in the locations of initial lineage establishment. Such effects could arise from among-state differences in the preferred destinations for travelers¹², in combination with year-to-year variability in where substantial circulation of viruses with high fitness occurs globally^{13,14,27,41} in the period when epidemic seeds would find fertile ground for transmission in the US. Testing this hypothesis and leveraging the predictive value that may be gained for prediction of epidemic composition underscores the need for strong global genomic and epidemiological surveillance networks.” (lines 491-498)